# A CSB-PAF1C axis restores processive transcription elongation after DNA damage repair

Diana van den Heuvel[1], Cornelia G. Spruijt[2,3,9], Román González-Prieto [4,9], Angela Kragten[1], Michelle T. Paulsen[5], Di Zhou[6], Haoyu Wu[1], Katja Apelt[1], Yana van der Weegen [1], Kevin Yang[5,7], Madelon Dijk[1], Lucia Daxinger[1], Jurgen A. Marteijn [6], Alfred C. O. Vertegaal[4], Mats Ljungman[5,8], Michiel Vermeulen [2] & Martijn S. Luijsterburg [1✉]

Bulky DNA lesions in transcribed strands block RNA polymerase II (RNAPII) elongation and induce a genome-wide transcriptional arrest. The transcription-coupled repair (TCR) pathway efficiently removes transcription-blocking DNA lesions, but how transcription is restored in the genome following DNA repair remains unresolved. Here, we find that the TCR-specific CSB protein loads the PAF1 complex (PAF1C) onto RNAPII in promoter-proximal regions in response to DNA damage. Although dispensable for TCR-mediated repair, PAF1C is essential for transcription recovery after UV irradiation. We find that PAF1C promotes RNAPII pause release in promoter-proximal regions and subsequently acts as a processivity factor that stimulates transcription elongation throughout genes. Our findings expose the molecular basis for a non-canonical PAF1C-dependent pathway that restores transcription throughout the human genome after genotoxic stress.

[1] Department of Human Genetics, Leiden University Medical Center, Leiden, The Netherlands. [2] Radboud Institute for Molecular Life Sciences, Oncode Institute, Radboud University Nijmegen, Nijmegen, The Netherlands. [3] Prinses Maxima Center, Utrecht, The Netherlands. [4] Department of Cell and Chemical Biology, Leiden University Medical Center, Leiden, The Netherlands. [5] Department of Radiation Oncology, University of Michigan, Ann Arbor, MI, USA. [6] Department of Molecular Genetics, Oncode Institute, Rotterdam, The Netherlands. [7] Department of Computational Medicine and Bioinformatics, University of Michigan, Ann Arbor, MI, USA. [8] Department of Environmental Health Sciences, University of Michigan, Ann Arbor, MI, USA. [9] These authors contributed equally: Cornelia G. Spruijt, Román González-Prieto. ✉email: m.luijsterburg@lumc.nl

The transcription of protein-coding genes involves RNA polymerase II enzymes (RNAPII), which pull DNA through their active sites and generate nascent transcripts. After initiation at the promoter, the majority of RNAPII molecules in metazoan cells pause at promoter-proximal sites, which is enforced by negative elongation factors DSIF and NELF[1,2]. The regulation of RNAPII pause release in response to environmental cues involves positive elongation factors, such as p-TEFb and the PAF1 complex (PAF1C)[1–3]. Both PAF1C and DSIF also act beyond pause release by stimulating the acceleration of RNAPII in promoter-proximal regions to ensure processive transcription elongation throughout genes[4–7].

The presence of bulky DNA damage in the transcribed strand of active genes is a major complication during transcription[8,9]. Persistent stalling of RNAPII at DNA lesions is highly toxic and constitutes an efficient trigger for apoptosis[10]. The presence of DNA lesions triggers a genome-wide transcriptional arrest due to stalling of elongating RNAPII at DNA lesions[8]. In addition, UV irradiation also inhibits transcription initiation through the stress-induced transcription repressor ATF3[11,12]. It is essential that cells overcome this arrest and restore transcription after repair to maintain gene expression.

The transcription-coupled nucleotide excision repair (TCR) pathway efficiently removes transcription-blocking DNA lesions through the sequential and cooperative recruitment of the ATP-dependent chromatin-remodeling factor CSB[13], the CUL4A-based (CRL4) E3 ubiquitin ligase complex containing CSA, and the UVSSA scaffold protein[14,15]. Mutations in the CSB and CSA genes cause Cockayne syndrome, which is characterized by severe developmental and neurological dysfunction[16,17]. In addition to protein–protein interactions, TCR complex assembly is tightly controlled by the CRL4$^{CSA}$-dependent ubiquitylation of RNAPII at a single lysine (K1268) of the largest RPB1 subunit[15]. The concerted action of CSB, CSA, UVSSA, and RPB1-K1268 ubiquitylation facilitate the association of the TFIIH complex with DNA damage-stalled RNAPII[14,15]. The subsequent association of XPA and XPG stimulate the translocase activity of TFIIH[18], likely resulting in TFIIH-mediated RNAPII displacement[19], which provides the endonucleases XPG and ERCC1-XPF access to excise the DNA lesion[20].

Although the TCR-mediated clearing of DNA lesions is essential, the precise mechanisms required for recovery of transcription after DNA repair remain unresolved. Stalled RNAPII molecules may be reactivated following TCR-mediated repair, which would require repositioning of the nascent transcript within the active site through hydrolysis to generate a new 3' end[21]. Alternatively, RNAPII may be released from the DNA template[22], followed by transcription recovery from the promoter[23]. Both CSA and CSB are essential for transcription recovery, which could be due to their role in clearing transcription-blocking DNA lesions[24]. In addition, the CS proteins also mediate the proteolytic degradation of ATF3 at later timepoints after UV irradiation, thereby eliminating its repressive impact on transcription initiation[11,12]. Furthermore, the histone chaperones HIRA[25] and FACT[26] and the histone methyltransferase DOT1L[27] play important roles in the recovery of transcription. However, the HIRA-dependent deposition of H3.3 and the FACT-mediated exchange of H2A at sites of local UV damage also occur in TCR-deficient cells[25,26]. Thus, the precise mechanisms involved in transcription recovery and their coordination with TCR-mediated repair remain to be established.

In this study, we define a new transcription recovery pathway that involves the CSB-dependent association of the PAF1 pausing and elongation complex with RNAPII specifically after UV irradiation. We show that PAF1 is dispensable for TCR-mediated repair, but specifically regulates RNAPII pause release and elongation activation from promoter-proximal regions. These findings identify a post-repair pathway that relies on CSB for the activation of paused RNAPII complexes by PAF1C to restore transcriptional activity and overcome DNA damage-induced silencing throughout the human genome.

## Results

**Identification of PAF1C as a UV-specific interactor of CSB.** To identify DNA damage-specific interactors of CSB, we stably expressed GFP-tagged CSB in SV40-immortalized CS1AN fibroblasts derived from a Cockayne syndrome B patient. Immunoprecipitation of GFP-CSB from the solubilized chromatin fraction of CS1AN-SV fibroblasts followed by SILAC-based mass spectrometry identified 172 proteins that showed at least 2-fold stronger association with chromatin-bound CSB isolated from UV-irradiated cells compared to undamaged cells. Among the top interactors were eight RNA polymerase II (RNAPII) subunits[28] and four polymerase-associated factor 1 complex (PAF1C) subunits[29] (Fig. 1a). To confirm these interactions in another cell line, we generated a CSB knockout (CSB-KO) in U2OS cells and subsequently re-expressed GFP-CSB in these cells. Expression of GFP-CSB rescued the sensitivity of CSB-KO cells to Illudin S, which causes transcription-blocking DNA lesions[30], confirming the functionality of the GFP-tagged CSB protein (Supplementary Fig. 1a). Label-free quantification proteomics after GFP-CSB pull-down confirmed a strong UV-induced association of GFP-CSB with RNAPII subunits, the CSA-DDB1-CUL4A complex, and all five PAF1C subunits (PAF1, LEO1, CTR9, WDR61, and CDC73; Fig. 1b). Intensity-based absolute quantification (iBAQ) of protein amounts[31], indicates that at least 60% of the isolated CSB molecules associate with PAF1C and RNAPII subunits after UV (Supplementary Fig. 1b). Co-IP experiments confirmed that CSB associated with RNAPII as well as with PAF1C subunits PAF1, LEO1, and CTR9 after UV irradiation (Fig. 1c, d) and treatment with Illudin S (Supplementary Fig. 1c).

**PAF1C associates with RNAPII and CSB after UV irradiation.** To further explore these interactions, we immunoprecipitated GFP-LEO1 from RPE1-hTERT cells and analyzed its interactome by both label-free mass spectrometry and western blot. GFP-LEO1 robustly interacted with PAF1, CTR9, WDR61, and CDC73 in both control and UV-exposed cells (Fig. 1e, f, Supplementary Fig. 1d). Both CSB and RNAPII strongly associated with GFP-LEO1 (Fig. 1e, f) only after UV irradiation, which was also observed after pull-down of transiently expressed GFP-CTR9 (Supplementary Fig. 1e). Based on the iBAQ values, we estimate that between 1 and 3% of the isolated LEO1 proteins associates with CS proteins and RNAPII subunits in response to UV irradiation, which is ~0.2% in unirradiated cells (Supplementary Fig. 1f, g).

Targeted immunoprecipitation and unbiased label-free proteomics on GFP-RPB1 revealed interactions with all eleven other RNAPII subunits and 22 Mediator subunits[32] in unirradiated cells, while only a very weak interaction with PAF1C subunits and no interaction with CSB was detected (Supplementary Fig. 1h). These findings demonstrate that PAF1C is a very low stoichiometric interactor of the RNAPII complex in undamaged cells. However, GFP-tagged RPB1 strongly associated with PAF1C subunits and CSB after UV as shown by label-free quantification proteomics (Fig. 1g) and western blot analysis (Fig. 1h). Based on the iBAQ values, we estimate that UV-induced DNA damage triggers a ~70-fold increase in the association between PAF1C and RNAPII, which is comparable with the UV-induced increase we detect between RNAPII and CSB (Supplementary Fig. 1i, j). Immunoprecipitation of endogenous RNAPII using either a

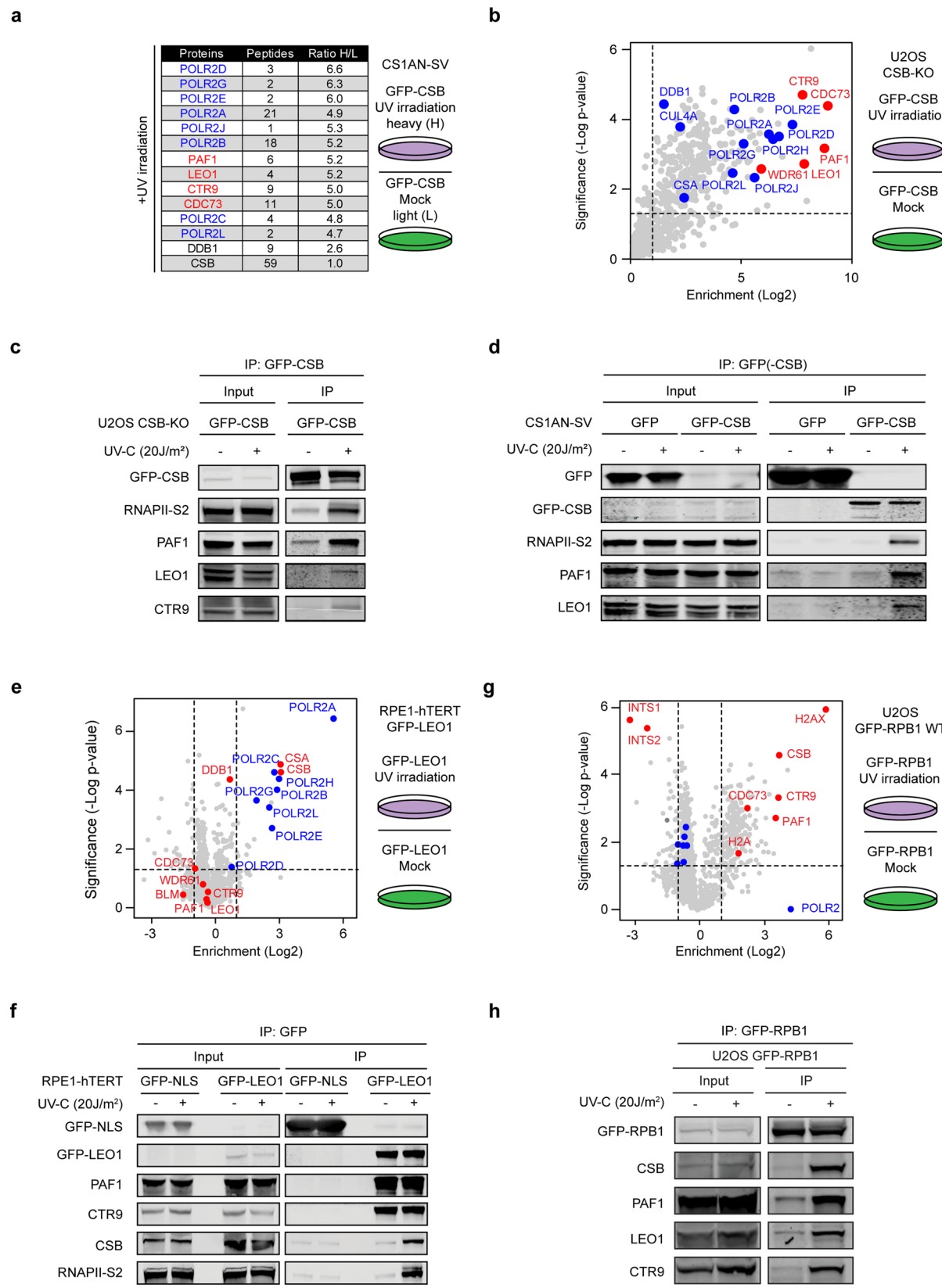

Ser5-P or a Ser2-P-specific antibody confirmed these interactions (Supplementary Fig. 1k).

**The UV-induced interaction between PAF1C and RNAPII is mediated by CSB.** The association of PAF1C with RNAPII

during the transcription cycle is fully dependent on the activity of CDK9 in the p-TEFb kinase complex[3,33]. To assess if the UV-induced interaction between PAF1C and RNAPII requires CDK9 activity, we treated cells with the selective CDK9 inhibitor LDC00067[34,35], which strongly reduced transcription (~70%) measured by incorporation of nucleotide analogue 5-ethynyl-

**Fig. 1 PAF1C is a UV-induced interactor of CSB. a** Results of SILAC-based MS after GFP-CSB pull-down from CS1AN-SV cells. The number of peptides identified and the UV-induced enrichment (ratio H/L) are shown. **b** Volcano plot depicting the UV-specific enrichment of proteins after pull-down of GFP-CSB from U2OS CSB-KO cells analyzed by label-free MS. The enrichment ($\log_2$) is plotted on the x-axis and the significance (two-sided t-test $-\log_{10}$ p-value) is plotted on the y-axis. Highlighted are significantly enriched subunits of RNAPII (blue) and PAF1C (red). **c** Co-immunoprecipitation of GFP-CSB from U2OS CSB-KO cells. **d** Co-immunoprecipitation of GFP or GFP-CSB from CS1AN-SV cells. **e** Volcano plot (as in **b**) depicting the enrichment of proteins or **f** co-immunoprecipitation after pull-down of GFP-LEO1 from RPE1-hTERT cells. **g** Volcano plot (as in **b**) depicting the enrichment of proteins or **h** co-immunoprecipitation after pull-down of GFP-RPB1 from U2OS cells. **c, d, f, h** All co-immunoprecipitation figures are representative examples of at least three independent experiments.

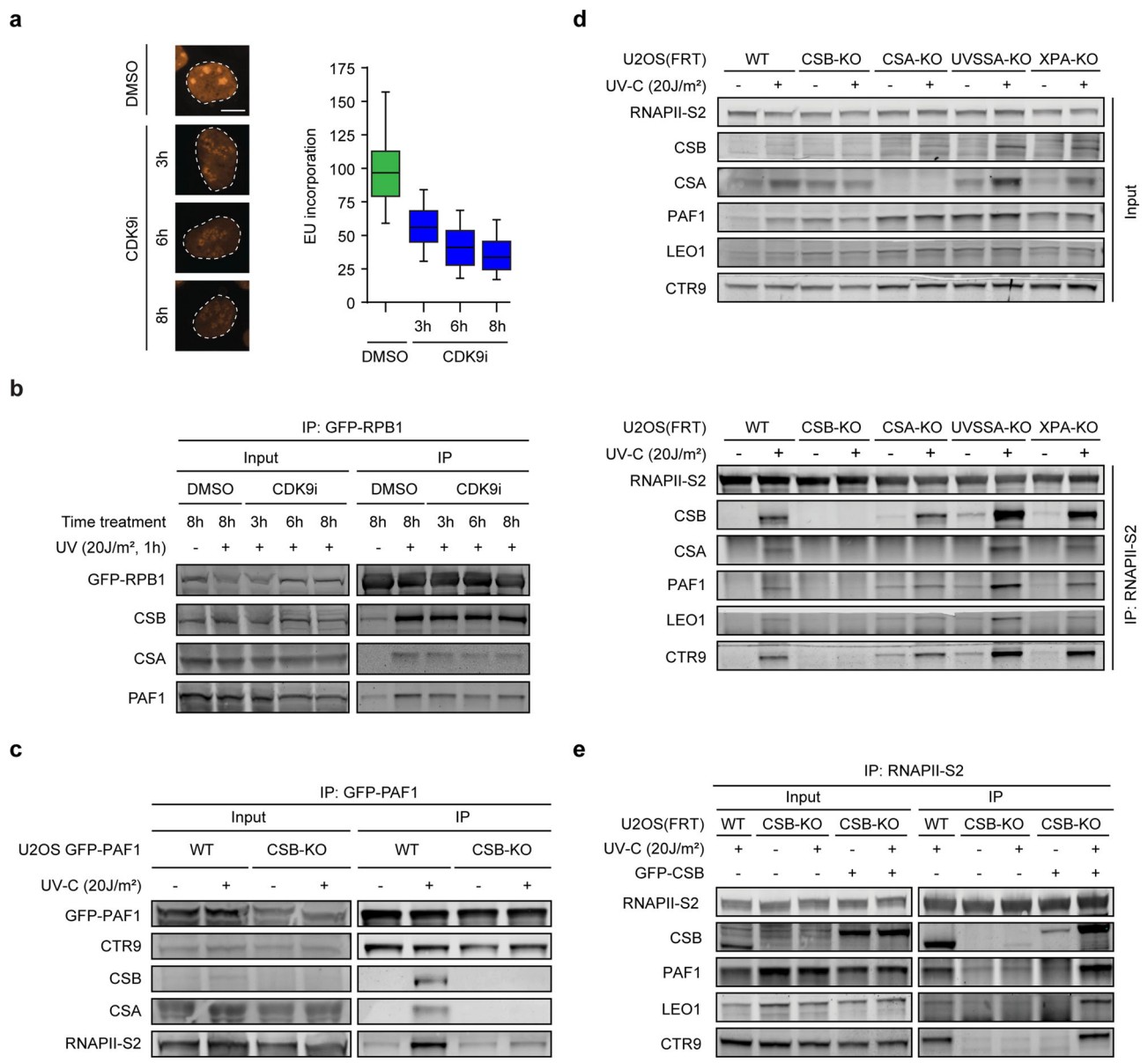

**Fig. 2 The RNAPII–PAF1C interaction is mediated by CSB. a** Representative images of U2OS cells treated with DMSO or 10 μM LDC00067 (CDK9 inhibitor), followed by pulse-labeling with 5-ethynyl-uridine (5-EU). Scale bar indicates 10 μm. Boxplots represent the median, 5th and 95th percentile of all cells of two independent experiments. **b** Co-immunoprecipitation of GFP-RPB1 in the presence of DMSO or 10 μM LDC00067 for the indicated time. **c** Co-immunoprecipitation of GFP-PAF1 from U2OS cells (WT or CSB-KO). **d** Co-immunoprecipitation of endogenous RNAPII-S2 from U2OS cells (WT or indicated KO). **e** As in **d**, but now including U2OS CSB-KO cells reconstituted with GFP-CSB. **b–e** All co-immunoprecipitation figures are representative examples of at least two independent experiments.

uridine (5-EU) in nascent transcripts (Fig. 2a). However, treatment with CDK9 inhibitor did not affect the UV-induced association between PAF1C and RNAPII (Fig. 2b), demonstrating that this UV-induced interaction is not mediated by the canonical p-TEFb-dependent pathway.

We next asked whether the interaction between PAF1C and RNAPII was dependent on CSB. While the UV-induced association of CSB, CSA, and RNAPII was detected after immunoprecipitation of GFP-PAF1, these interactions were abolished in CSB-KO cells without affecting the constitutive

association of GFP-PAF1 with CTR9 (Fig. 2c). Endogenous PAF1C still associated with RNAPII after UV in cells knockout of CSA, UVSSA[14], or XPA (Fig. 2d, Supplementary Fig. 2a) demonstrating that this loss of interaction is not due to a general TCR deficiency. Importantly, all TCR knockout cells were highly sensitive to Illudin S, which was fully rescued by stable re-expression of the corresponding TCR protein (Supplementary Fig. 2b). To validate these interactions, we generated knockouts of CSA, CSB and UVSSA in U2OS cells stably expressing GFP-RPB1. The knockout of these genes was confirmed by sequencing, western blot analysis, and Illudin S survival experiments (Supplementary Fig. 2c, d, e). Immunoprecipitation of GFP-RPB1 confirmed that knockout of CSB, but not other TCR genes, prevented the UV-induced PAF1C–RNAPII interaction (Supplementary Fig. 2e), which was confirmed by label-free proteomics (Supplementary Fig. 2f). Similar results were obtained after pull-down of Ser5-P-modified RNAPII (Supplementary Fig. 2g). Re-expression of GFP-CSB fully restored the association between RNAPII and PAF1C in CSB-KO cells after UV (Fig. 2e), establishing that CSB mediates the UV-induced interaction between PAF1C and RNAPII.

**The UV-induced interaction between RNAPII and CSB is stabilized by PAF1**. To better understand how the interactions between CSB, PAF1C, and RNAPII might be mediated, we first compared available cryo-EM structures for their interactions. The yeast orthologue of CSB, RAD26, is bound behind RNAPII to the upstream DNA that has just been transcribed[36] (Fig. 3a). Interestingly, the human PAF1C complex is bound to the outer surface of RNAPII with the central CTR9 subunit docking onto the polymerase funnel, while the C-terminus of the LEO1 subunit contacts the upstream DNA that has just been transcribed[33] (Fig. 3a). Based on this comparison, we postulate that LEO1 may have direct protein–protein contacts with CSB. To test this possibility, we stably expressed GFP-PAF1$^{WT}$ and a GFP-tagged mutant of PAF1 lacking five amino acids required for its association with LEO1 (PAF1$^{\Delta LEO1}$)[37]. While GFP-PAF1$^{WT}$ interacted robustly with both CTR9 and LEO, we found that GFP-PAF1$^{\Delta LEO1}$ interacted normally with CTR9, but failed to associate with LEO1 (Fig. 3b). Interestingly, PAF1$^{\Delta LEO1}$ interacted much less efficiently with CSB, CSA, and RNAPII after UV irradiation compared to PAF1$^{WT}$, suggesting that the LEO1 subunit likely anchors the PAF1C complex to CSB and RNAPII after UV (Fig. 3b).

Our interaction experiments revealed that CSB is essential to mediate the association of PAF1C with RNAPII after UV (Fig. 2c–e). We next asked whether the reverse is also true and if PAF1 is required for CSB to associate with RNAPII after UV. To test this, we knocked in an auxin-inducible degron (AID) into both alleles of the endogenous *PAF1* locus in U2OS cells expressing the rice-specific F-box gene *TIR1*[38] (Supplementary Fig. 3a). Treatment of knockin cells with auxin led to a strong depletion of PAF1, which was accompanied by reduced protein levels of CTR9 (Fig. 3c, Supplementary Fig. 3b). Pull-down experiments of RNAPII-S2 at multiple timepoints after UV irradiation revealed that CSB still interacted with RNAPII after UV, and that UV-induced ubiquitylation of RPB1 could also be detected in PAF1-depleted cells with similar kinetics as in TIR1 control cells (Fig. 3c). However, the amount of CSB and ubiquitylated RNAPII were slightly reduced in PAF1-depleted cells, suggesting that PAF1C is not essential for, but may stabilize the interaction between CSB and RNAPII after UV (Fig. 3c).

**PAF1C is not required for clearing DNA lesions by TCR**. We next sought to address if PAF1C has a direct role in DNA repair.

The majority of UV-induced DNA lesions throughout the genome is removed by global-genome repair (GGR), while a small subset of lesions in actively transcribed strands is eliminated by TCR. To rule out a role of PAF1 in GGR, we measured unscheduled DNA synthesis (UDS) after local UV irradiation by pulse-labeling with the nucleotide analogue 5-ethynyl-deoxy-uridine (EdU). Robust EdU incorporation could be detected in UV-irradiated TIR1 control cells, which was strongly suppressed by knockdown of XPA (Supplementary Fig. 3c). However, deletion of PAF1 by treatment with auxin did not affect UV-induced EdU incorporation (Supplementary Fig. 3c), ruling out a role of PAF1 in GGR.

To specifically capture TCR-mediated repair, we employed nondividing primary XP-C patient-derived fibroblasts, which are deficient in GGR (Fig. 3d–f). These cells were globally irradiated with UV-C light (8 J/m$^2$) and pulse-labeled for 8 h with EdU. TCR-specific UDS was visualized using Click-It chemistry combined with tyramide-based signal amplification[39]. Robust incorporation of EdU was detected in UV-irradiated XP-C cells, but not in unirradiated controls cells (Fig. 3d–f). Knockdown of CSB with specific siRNAs prevented the incorporation of EdU after UV. Importantly, knockdown of PAF1 with two independent siRNAs (Fig. 3d), did not affect TCR-specific repair synthesis (Fig. 3e, f), demonstrating that PAF1 has no direct role in TCR.

**PAF1C promotes transcription recovery after UV irradiation**. The presence of UV-induced DNA lesions triggers a strong transcription arrest. To address whether PAF1C plays a role in the recovery of transcription after repair, we visualized nascent transcription by 5-ethynyl-uridine labeling following global UV irradiation after knockdown of TCR proteins or PAF1 (Fig. 4a–c). Nascent transcription was strongly inhibited at 3 h after UV irradiation in all conditions (Fig. 4b–e). Significant transcription recovery was detected at 18 h after UV in controls cells, but not in TCR-deficient XPA knockdown cells (Fig. 4b–e), or CSA knock-out cells (Supplementary Fig. 3d). Knockdown of PAF1 with two independent siRNAs significantly impaired the ability of cells to recover transcription following UV irradiation (Fig. 4b, d, e), which could be reversed by re-expression of siRNA-resistant GFP-PAF1 (Fig. 4c, d, e).

To validate these findings, we also performed these experiments in the PAF-AID knockin cells following auxin treatment. Single knockin clones showed strong auxin-induced depletion of PAF1 within 5 h (Fig. 4f). Visualizing nascent transcription in two independent PAF1-AID clones revealed a failure to restore transcription after UV, while TIR1 control cells showed full transcription recovery (Fig. 4g, h). These findings uncover an important role of PAF1C in transcription recovery following genotoxic insult. Importantly, knockdown of XPA in either TIR1 control cells or PAF1-depleted cells impaired transcription recovery to the same extent (Supplementary Fig. 3d), suggesting that TCR and PAF1-mediated transcription restart operate in the same pathway. Considering that PAF1 is not directly involved in TCR (Fig. 3f), our findings suggest that PAF1-mediated transcription restart occurs after the elimination of transcription-blocking DNA lesions by TCR.

**Genome-wide repositioning of PAF1 at the TSS after UV irradiation**. Transcription recovery after repair may involve the activation of paused RNAPII from promoter-proximal regions in a PAF1-stimulated manner[3,5,33,40,41]. To address this possibility, we mapped PAF1 chromatin-binding sites in the genome using ChIP sequencing (ChIP-seq)[42] in the absence of DNA damage and 8 h after UV irradiation, when transcription starts to recover and CSB–RNAPII–PAF1C interactions still take place (Fig. 5a). PAF1 was bound predominantly to transcription start sites (TSSs)

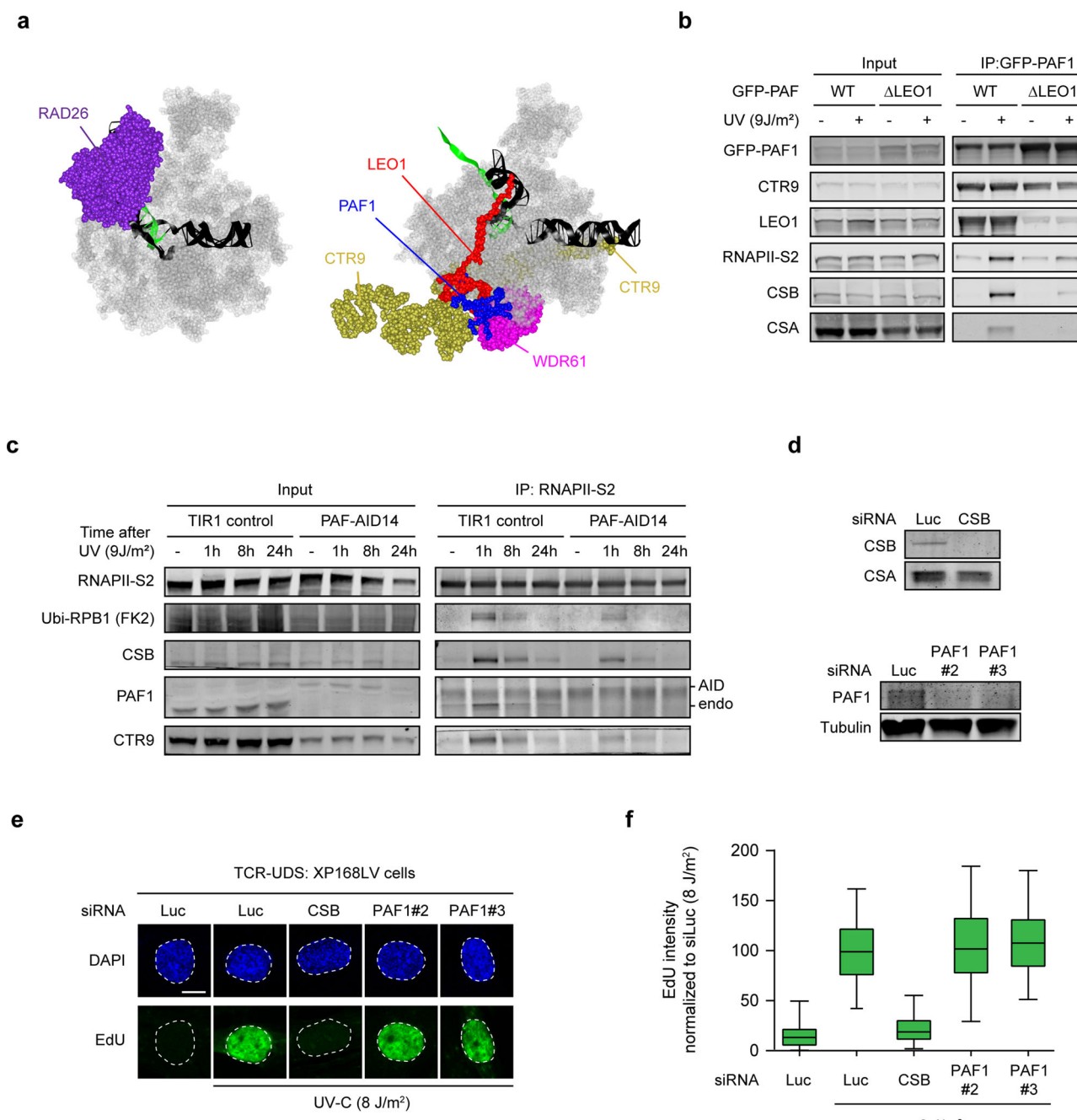

**Fig. 3 PAF1C stabilizes the RNAPII–CSB interaction, but is not involved in TCR. a** The left panel shows the Cryo-EM structure of RNAPII (in silver) bound to RAD26, which is the yeast orthologue of CSB, from the Wang lab (PDB-ID: 5VVR). The right panel shows the Cryo-EM structure of RNAPII (in silver) bound to PAF1C subunits (color-coded as indicated in the figure) from the Cramer lab (PDB-ID: 6GMH). **b** Co-immunoprecipitation of GFP-PAF1 in cells expressing either GFP-PAF1$^{WT}$ or GFP-PAF1$^{\Delta LEO1}$. **c** Co-immunoprecipitation of endogenous RNAPII-S2 in U2OS TIR1 cells or U2OS PAF-AID cells after depletion of PAF1 and at indicated timepoints after UV. **d** Validation of siRNA-mediated knockdown of CSB or PAF1 in XP168 LV cells by western blot analysis. Representative figure of two replicates. **e** TCR-UDS assay in XP-C primary fibroblasts (XP168LV) during 8 h following UV in cells transfected with the indicated siRNAs validated in **d**. Scale bar indicates 10 μm. **f** Quantification of TCR-UDS signal from **e**. Boxplots represent the median, 5th and 95th percentile of all cells of at least two independent experiments. **b, c** All co-immunoprecipitation figures are representative examples of at least two independent experiments.

and downstream from transcription termination sites (TTSs) (Fig. 5b, c, Supplementary Fig. 4a, b). Heatmaps of the distribution of reads around TSS sites revealed a large degree of overlap between our and recently published PAF1 ChIP-seq data[41]. From the set of 8811 genes, we detect robust binding of PAF1 to the TSS and downstream from the TTS in a subset of ~3000 genes (Fig. 5d, Supplementary Fig. 4b, c). The top 3000 genes from our

analysis were also bound by CSB and became most strongly bound by CSA and ATF3 after UV (Supplementary Fig. 4d), suggesting that these interactions might all take place at the TSS, and that these genes are subjected to ATF3-mediated transcriptional repression after UV[11].

PAF1 became more restricted to the TSS region and showed substantially reduced binding in gene bodies and downstream of

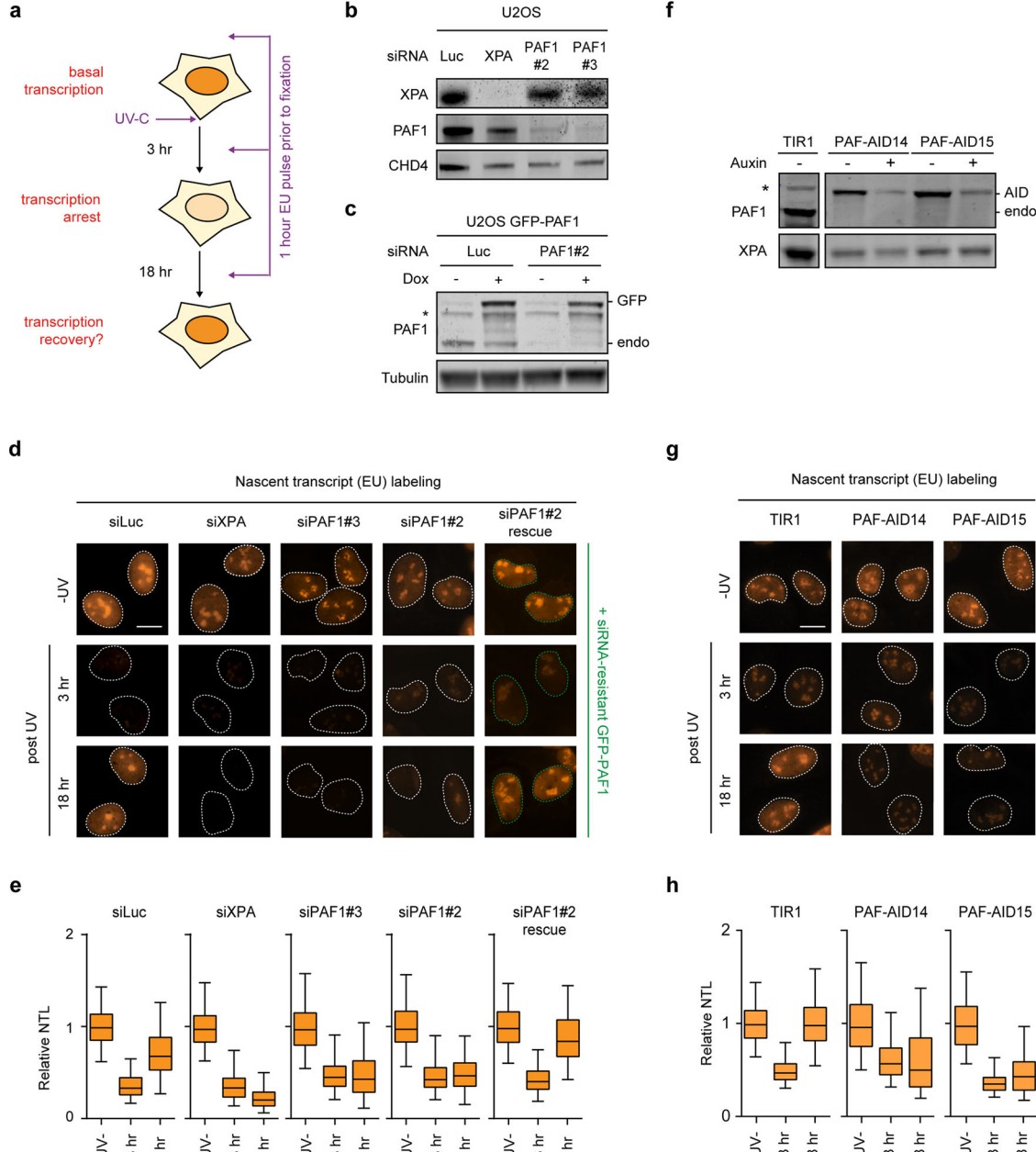

**Fig. 4 PAF1C loss impairs transcription recovery after UV irradiation. a** Experimental outline of the nascent transcription analyses. **b** Validation of the knockdown of XPA and PAF1 in U2OS cells by western blot analysis. **c** Validation of the knockdown of endogenous PAF1 in U2OS cells ectopically expressing siRNA-resistant PAF1 by western blot analysis. **d** Representative images of U2OS cells transfected with the indicated siRNAs after pulse-labeling with 5-ethynyl-uridine (5-EU). Cells with green outlines express GFP-tagged siRNA-resistant PAF1. Scale bar indicates 10 μm. **e** Quantification of nascent transcript levels (NTL) from **d**. Boxplots represent the median, 5th and 95th percentile of all cells of three independent experiments. **f** Validation of auxin-induced degradation of endogenous PAF1 in two independent U2OS PAF1-AID knockin clones. **g** Representative images of auxin-treated U2OS TIR1 control or two PAF1-AID clones after pulse-labeling with 5-ethynyl-uridine (5-EU). Scale bar indicates 10 μm. **h** Quantification of nascent transcript levels from **g**. Boxplots represent the median, 5th and 95th percentile of all cells of four independent experiments. **b, c, f** Representative figure of at least two replicates.

the TTS at 8 h after UV irradiation with both 6 J/m$^2$ and 9 J/m$^2$ (Supplementary Fig. 4a, b, e). At this time-point, cells already start to resume transcription, but have not yet fully completed transcription recovery[23]. Immunoprecipitation of RNAPII indeed showed a strong interaction with PAF1C subunits at 8 h after UV, which was lost at 24 h after UV when cells have fully recovered transcription (Fig. 5a). Strikingly, we detected a marked shift in PAF1 binding by ChIP-seq away from the promoter into the first ~1 kb downstream of the TSS at 8 h after UV, which was no

longer observed at 26 h after UV (Fig. 5e). Interestingly, since PAF1C is an important regulator of RNAPII pause release[3,33,40,41], our data might suggest a role for PAF1C in releasing RNAPII from pause sites in a CSB-dependent mechanism to restore transcription after UV irradiation.

**A genome-wide shift of RNAPII into gene bodies after UV irradiation.** To explore this possibility further, we mapped

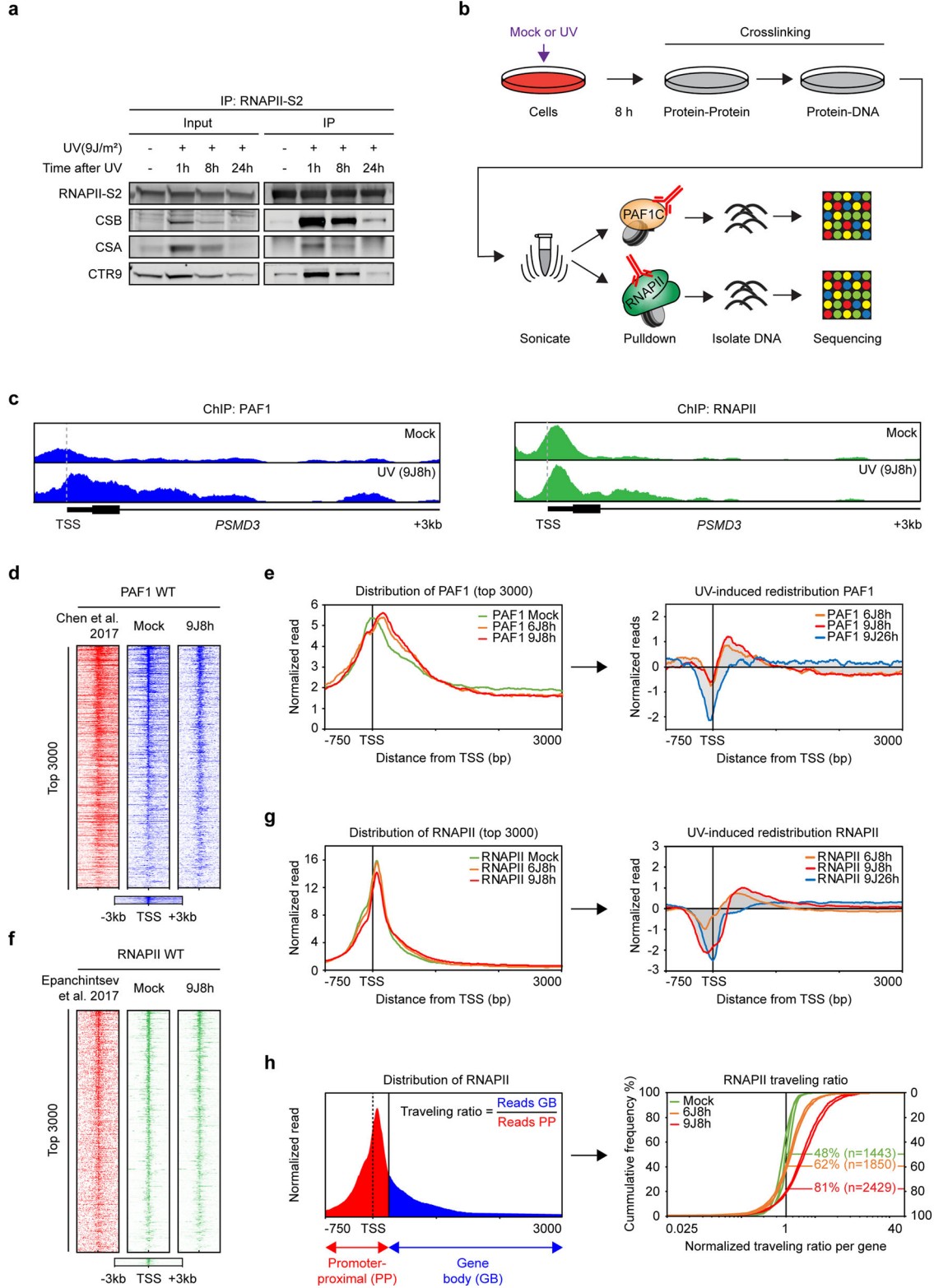

RNAPII-binding sites by ChIP-seq. RNAPII most strongly associated with the TSS as well as downstream of the TTS in the same subset of 3000 genes that were also bound by PAF1 with a high degree of overlap with published RNAPII ChIP-seq data (Fig. 5c, f, Supplementary Fig. 5a)[11]. UV irradiation triggered a dose-dependent release of RNAPII into the first ~2 kb downstream of the TSS at 8 h after UV irradiation (Fig. 5g), which coincided with the region to which a UV-induced shift in PAF1 binding was also observed (Fig. 5e). The shift in RNAPII binding was no longer detected at 26 hrs after UV irradiation (Fig. 5g, Supplementary Fig. 5b), coinciding with the loss of PAF1C binding to RNAPII (Fig. 5a), and near complete transcription recovery (Fig. 4e, h). Representative examples of short (ARF6, 5 kb), intermediate (NDUFS5, 10 kb) and longer genes (PSMD3, 24 kb) all show the UV-induced redistribution of both PAF1 and RNAPII within the first 2 kb of the gene (Supplementary Fig. 5c).

**Fig. 5 ChIP-seq reveals UV-induced repositioning of PAF1 and RNAPII into promoter-proximal regions. a** Co-immunoprecipitation of endogenous RNAPII-S2 from U2OS cells at different timepoints after UV. Note that the input sample was taken after chromatin fractionation and therefore show chromatin-bound protein levels rather than whole cell protein levels, which are equal between all conditions. Representative figure of at least three independent experiments. **b** Outline of the ChIP-seq approach to map PAF1- and RNAPII-binding sites. **c** UCSC genome browser track showing the read density of (left) PAF1 and (right) RNAPII signal across the *PSMD3* gene in unirradiated and UV-irradiated cells. Tracks represent pooled reads of 3 mock and 3 9J8h PAF1 ChIP-seq replicates and 3 mock and 2 9J8h RNAPII ChIP-seq replicates. **d** Representative heatmaps from PAF1 ChIP-seq data around the transcription start sites (TSS) of the top 3000 genes that bind PAF1. Data is ranked based on the PAF1 signal in unirradiated cells (mock; in blue), and compared to published PAF1 ChIP-seq data (in red), and PAF1 ChIP-seq at 8 h after 9 J/m$^2$ UV irradiation (in blue; 9J8h). **e** Averaged metaplots of PAF1 ChIP-seq of the top 3000 genes around the TSS in unirradiated (mock; $n = 3$) and UV-irradiated cells (8 h after 6 J/m$^2$ ($n = 1$) and 9 J/m$^2$ ($n = 3$)). The right panel shows the UV-induced redistribution of PAF1 calculated by subtracting the mock from the +UV distribution profiles for 6 J/m$^2$ at 8 h and for 9 J/m$^2$ at 8 h and 26 h ($n = 1$). **f** Representative heatmaps from RNAPII ChIP-seq data around the TSS of genes as in **d**. Data of unirradiated cells (mock; in green) are compared to published RNAPII ChIP-seq data (in red), and RNAPII ChIP-seq at 8 h after 9 J/m$^2$ UV irradiation (in green; 9Jh8). **g** As in **e**, but for RNAPII with averages of mock ($n = 3$), 8 h after 6 J/m$^2$ ($n = 3$) and 8 h ($n = 2$) or 26 h ($n = 2$) after 9 J/m$^2$. **h** Schematic representation of the traveling ratio of RNAPII, which is calculated by dividing the reads of the gene body (+250 bp to +3 kb; blue) over the reads in the promoter-proximal region (−750bp to +250 bp; red). The right panel shows the ratio of the RNAPII traveling ratio (or the normalized traveling ratio) for 3000 genes relative to the average traveling ratio in the unirradiated control (set to 1). Shown are three independent replicates in unirradiated cells (mock, green), and three replicates after UV irradiation with 6 J/m$^2$ (in orange) and two replicates after 9 J/m$^2$ (in red). The *y*-axes indicate percent of all genes. Percentages and *n* indicated in the plot refer to the percentage and number of the 3000 genes with a normalized traveling ratio above 1.

To further quantify the release of RNAPII after UV, we calculated the traveling ratio for each gene, which was defined as the density of RNAPII reads within the first 3 kb of the gene body relative to the density in the promoter-proximal region (Fig. 5h). An increased traveling ratio indicates that more RNAPII molecules have shifted into the gene body. This approach revealed a dose-dependent increase of the normalized traveling ratio after UV (Fig. 5h). Importantly, independent repeats of these different genome-wide ChIP-seq conditions showed a highly similar shift (Fig. 5h, Supplementary Fig. 6, Supplementary Table 6), suggesting that these changes in RNAPII binding across the genome reflect a very robust cellular response.

**The release of RNAPII after UV irradiation is dependent on PAF1.** We next quantified the UV-induced repositioning of PAF1 binding in individual genes by calculating the ratio of reads in the first 1 kb of genes relative to the 750 bp upstream of the TSS in wild-type cells (Fig. 6a, b see the *POLG2* and *SLC40A1* genes in Fig. 6c). Strong and consistent UV-induced repositioning of PAF1 was detected in 478 of the 3000 genes at 8 h after UV in all 6 J/m$^2$ and 9 J/m$^2$ replicates in wild-type cells (Fig. 6d). The remaining 2522 of the 3000 genes showed no repositioning of PAF1 downstream of the TSS in at least one of the replicates in wild-type cells (Fig. 6d). Notably, we observed a stronger RNAPII shift into gene bodies after UV in the set of 478 genes that displayed consistent PAF1 repositioning compared to the set of 2522 genes without PAF1 repositioning (Fig. 6e). Thus, the release of RNAPII after UV irradiation correlates strongly with the repositioning of PAF1.

To further address whether PAF1 is needed for the UV-induced release of RNAPII, we performed ChIP sequencing in TIR1 control cells and PAF1-AID knockin cells. The RNAPII ChIP-seq profiles in PAF1-depleted human cells were very similar to published data in mouse cells[5] (see our reanalysis of these data in Supplementary Fig. 7a, b). TIR control cells showed a dose-dependent shift of RNAPII after UV irradiation in the top 3,000 genes (Fig. 6f), similar to our data in U2OS cells (Fig. 5h). Auxin-induced depletion of PAF1 strongly prevented the shift of RNAPII in many of the 3000 genes (Fig. 6f). Even when we selected the 902 genes with the strongest shift in RNAPII in the TIR1 control cells, we could not detected an appreciable shift in PAF1-depleted cells (Fig. 6g). These findings show that PAF1 stimulates the UV-induced repositioning of RNAPII around promoters.

**CSB stimulates PAF1 release and Ub-H2B deposition in a subset of genes after UV.** We next asked whether the repositioning of PAF1 away from TSS sites is dependent on CSB. To address this, we mapped chromatin-binding sites of PAF1 by ChIP-seq in CSB knockout (KO) cells (Fig. 7a, b). In unirradiated cells, we did not detect major differences in PAF1 binding to promoter regions in the genomes of wild-type or CSB-KO cells (compare Fig. 5d, e to Fig. 7a, c).

Importantly, the UV-induced repositioning of PAF1 after UV in the top 3000 genes in wild-type cells (Fig. 5e) was less prominent in CSB-KO cells (Fig. 7b, c). A subsequent analysis focusing on a subset of 478 genes with a consistent UV-induced repositioning of PAF1 in wild-type cells (Fig. 6b), revealed that 306 of these genes no longer showed a UV-induced shift of PAF1 in CSB-KO cells (Fig. 7d, e). These findings reveal that CSB stimulates the UV-induced repositioning of PAF1 in a subset of genes.

To investigate how the inability to reposition PAF1 affects the UV-induced release of RNAPII, we also mapped chromatin-binding sites of RNAPII by ChIP-seq in CSB-KO cells (Fig. 7f). While loss of CSB did not appreciably affect the binding of RNAPII to TSS sites compared to wild-type cells without irradiation (compare mock in Fig. 7f to Fig. 5f), we detected strongly reduced binding of RNAPII at 8 hrs after UV in CSB-KO cells (Fig. 7f, g), which was not observed under similar conditions in wild-type cells (Fig. 7h). These findings suggest that CSB-KO cells show a strong transcriptional repression at 8 hrs after UV irradiation, through a combination of defective TCR, and their inability to activate PAF1 and remove repressor ATF3 from TSS sites[11].

The ubiquitylation of H2B (Ub-H2B) is a co-transcriptionally deposited histone mark that correlates with RNAPII elongation rates and PAF1C activity[5,43]. We therefore decided to map Ub-H2B deposition by ChIP-seq in wild-type and CSB-KO cells after UV irradiation. We detected clear Ub-H2B levels throughout genes in both unirradiated wild-type and CSB-KO cells (see Fig. 7i for the *KANSL1* gene). At 8 h after UV in wild-type cells, Ub-H2B levels were similar throughout early gene bodies (< 20 kb), with decreasing levels toward the end of longer genes (>100 kb). This is consistent with ongoing, but incomplete recovery of RNA synthesis particularly in long genes in wild-type cells at this time-point[23]. Strikingly, in UV-irradiated CSB-KO cells, Ub-H2B levels progressively decreased much more rapidly within the first 20 kb of genes (see Fig. 7i for the *KANSL1* gene). We confirmed by immunofluorescence that absolute Ub-H2B levels were strongly

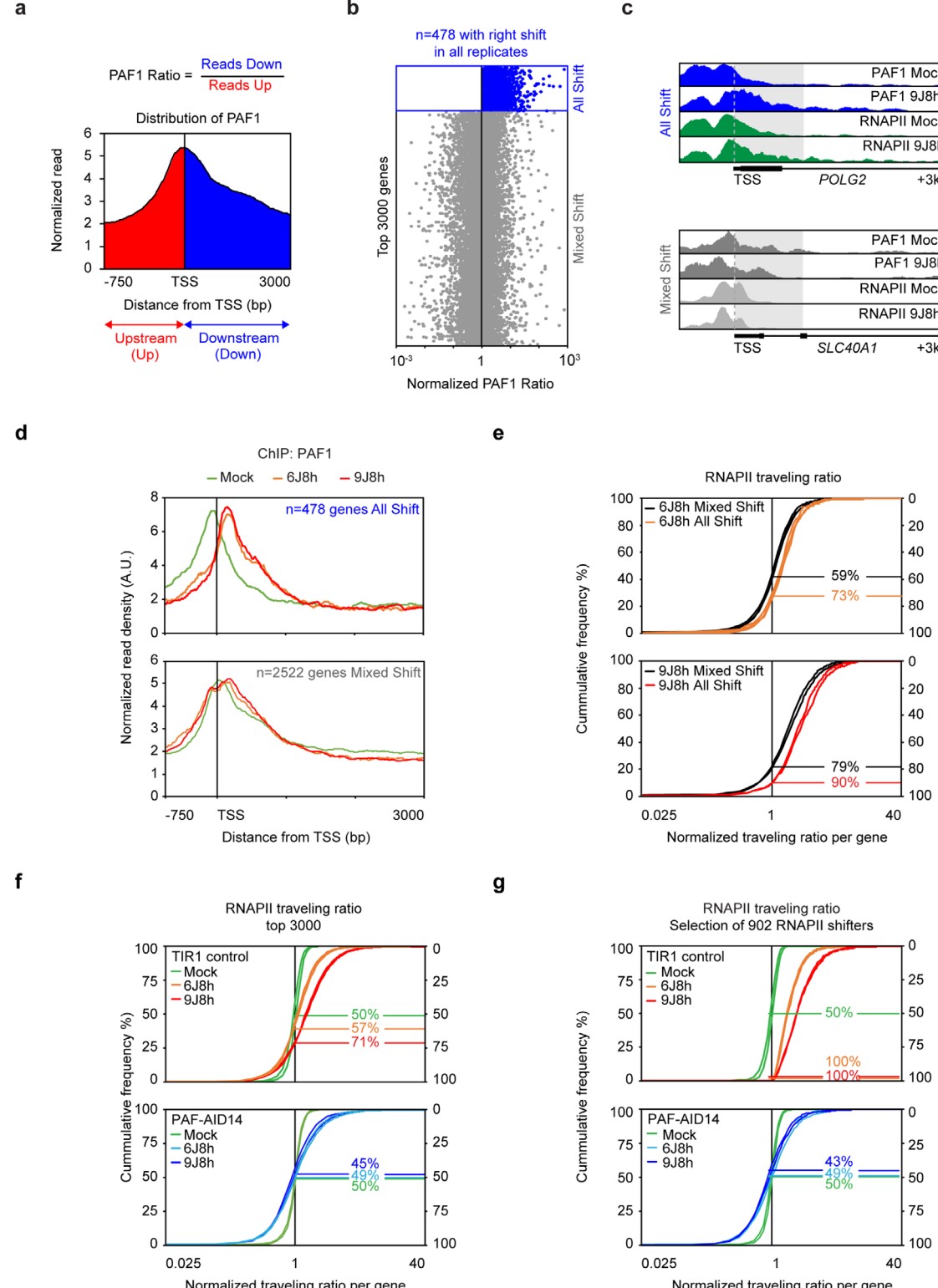

decreased in CSB-KO cells (38%) compared to wild-type cells (63%) at 8 h after UV irradiation (Fig. 7j). Metaplots of Ub-H2B averaged over 820 genes revealed a faster and more progressive loss of Ub-H2B deposition in CSB-KO cells in gene bodies after UV irradiation compared to wild-type cells (Fig. 7k).

**PAF1 stimulates processive transcription elongation after UV irradiation**. During transcription regulation, PAF1C travels with

RNAPII and promotes efficient elongation through chromatin[5,6]. To measure transcription elongation after UV, we metabolically pulse-labeled nascent transcripts with bromouridine (BrU) at different timepoints after UV irradiation, followed by capture and sequencing of the BrU-labeled nascent RNAs (Fig. 8a)[23]. Nascent transcription was substantially reduced at TSS sites and progressively decreased further into gene bodies at 3 h after UV in control cells (Fig. 8b–d, see Supplementary Fig. 7c for an independent replicate)[23,44]. This is

**Fig. 6 The UV-induced repositioning of RNAPII into promoter-proximal regions requires PAF1C. a** Schematic representation of the traveling ratio of PAF1, which is calculated by dividing the reads of the gene body (TSS to +1000 bp; blue) over the reads in the promoter-proximal region (−750bp to TSS; red). **b** Quantification of the traveling ratio (or shift) in PAF1 binding after UV. All replicates and UV doses (6 and 9 J/m$^2$, total $n = 4$) were pooled, which revealed a set of 478 genes that shows a uniform UV-induced shift into promoter-proximal regions in all replicate experiments at 8 h after 6 and 9 J/m$^2$ (All Shift) and a set of 2522 genes that show a UV-induced shift only in a subset of the conditions (Mixed Shift). **c** UCSC genome browser track showing the read density of PAF1 and RNAPII signal across the *POLG2* gene (All Shift) and *SLC40A1* gene (Mixed Shift) in unirradiated (mock) and UV-irradiated cells. **d** Averaged metaplots of PAF1 ChIP-seq around the TSS of the 478 All Shift genes (upper panel) or the 2522 Mixed Shift genes (lower panel) in unirradiated cells (mock, $n = 3$) and 8 h after UV with 6 J/m$^2$ ($n = 1$) and 9 J/m$^2$ ($n = 3$). **e** The ratio of the RNAPII traveling ratios for the 478 All Shift genes compared to the 2522 Mixed Shift genes at 8 h after UV with 6 J/m$^2$ ($n = 3$) and 9 J/m$^2$ ($n = 2$). **f** The ratio of the RNAPII traveling ratios for 3000 genes relative to the average traveling ratio in the unirradiated control (set to 1) shown for TIR1 (upper panel) or PAF1-AID clone 14 (lower panel). Shown are unirradiated cells (mock, $n = 2$), 8 h after 6 J/m$^2$ (6J8h, $n = 2$), or 9 J/m$^2$ (9J8h, $n = 2$). **g** Data as in **f**, but for 902 genes with consistent shift of RNAPII in all replicates at 6 J/m$^2$ or 9 J/m$^2$ compared to mock in TIR1 cells. The y-axes in panels **e**–**g** indicate percent of all genes. Indicated percentages and n indicate genes with a normalized traveling ratio above 1.

consistent with the distribution of DNA lesions in transcribed strands after UV irradiation with 7 J/m$^2$ (1 CPD/16 kb[45]), and reflects the probability of RNAPII molecules encountering a DNA lesion. A partial restoration of reads at the TSS and from within gene bodies in TIR1 cells could already be detected at 8 h after UV, and this was fully restored at 24 h after UV (Fig. 8b–d, Supplementary Fig. 7c). This restoration was accompanied by the reappearance of both RNAPII and PAF1 at TTS sites detected by ChIP-seq (see the *ZFR* gene in Fig. 8d). These data suggest that transcription recovery occurs in a wave starting from the promoter-proximal region and ultimately reaching the end of genes.

Unirradiated cells depleted for PAF1 showed normal nascent transcription in the first 50 kb and slightly reduced nascent transcription toward the end of long genes (Fig. 8b, Supplementary Fig. 7c, d). This is in line with a role of PAF1 as elongation factor[5], but also shows that under our experimental conditions there is no dramatic impact of PAF1 depletion on general transcription (Supplementary Fig. 7d). Following UV irradiation, PAF1-depleted cells showed a strong and progressive loss of reads into gene bodies at 3 h after UV irradiation, which was not restored at both 8 and 24 h after UV (Fig. 8b–d, Supplementary Fig. 7c, d). This effect was reminiscent of the progressive loss of Ub-H2B deposition in CSB-deficient cells at 8 h after UV (see Fig. 8d for the *ZFR* gene). The impact of PAF1 depletion on nascent transcription was most striking for long genes (>100 kb), but also observed in shorter genes after UV (Fig. 8b). In contrast to control cells, PAF1-depleted cells did not show reduced transcription in promoter-proximal regions at 3 and 8 h after UV (Fig. 8b), which coincided with increased RNAPII occupancy in this region detected by ChIP-seq (Supplementary Fig. 7a, b)[5]. Thus, PAF1-depleted cells may accumulate aberrant prematurely terminated transcripts in the TSS region early after UV (Fig. 8b), while nascent transcription is strongly decreased at 24 h after UV (Fig. 8b, Supplementary Fig. 7d). Although limited transcription initiation and/or pause release may still be possible without PAF1 at 24 h after UV, these RNAPII molecules are not activated for productive and processive elongation and do not make it to the end of genes (Fig. 8b–d, Supplementary Fig. 7d).

## Discussion

Under undamaged conditions, PAF1C is known to interact with RNAPII in a p-TEFb-dependent manner to regulate transcription elongation[3,5,33]. Our protein–protein interaction data, however, suggests that in unirradiated cells only a small RNAPII pool stably interacts with the PAF1C and that UV irradiation strongly increases this pool in a manner that does not require the canonical p-TEFb pathway (Figs. 1–2). Instead this UV-induced PAF1C–RNAPII interaction relies on the TCR-specific CSB protein (Figs. 1–2). The RNAPII–PAF1C interaction still occurred normally in TCR-deficient *CSA* or *UVSSA* knockout cells,

suggesting that the loss of interaction in CSB-KO cells is not due to a general TCR deficiency (Fig. 2). It is possible that CSB mediates the association of PAF1C to RNAPII through protein–protein interactions with LEO1, which is compatible with available structural data[33] (Fig. 3a), and supported by our interaction data (Fig. 3b). The fact that we only detect a small RNAPII pool that stably interacts with PAF1C in unirradiated cells (Figs. 1 and 2) could be explained by the presence of a small pool of p-TEFb-modified RNAPII at any given time, or by a transient interaction that is not captured efficiently under our experimental IP conditions. Such a transient interaction may be stabilized by DNA-protein crosslinking applied in ChIP-seq experiments, explaining why we do detect both RNAPII and PAF1 at the TSS of the top 3000 genes in our study (Figs. 5 and 6).

During the canonical transition of paused RNAPII into productive elongation, the association of PAF1C with RNAPII displaces the NELF complex from RNAPII. Interestingly, a p38 MAP kinase pathway releases NELF from chromatin after UV irradiation, which is partially dependent on CSB[46]. It is therefore possible that CSB could promote efficient NELF displacement by regulating both PAF1C recruitment and activating the p38 pathway in parallel. In addition to NELF release, p38 also activates a p-TEFb pathway that regulates a transcriptional response immediately after DNA damage induction. This pathway ensures the expression of short coding and noncoding RNAs involved in the DNA damage response, including *FOS* and *CDKN1A*[47]. Importantly, many of these short DDR genes are regulated by p53 and do not require CSB for their expression after UV irradiation[48]. Thus, cells mount an immediate transcriptional response through p-TEFb to ensure expression of short DDR genes and noncoding RNAs while most other genes undergo a transcriptional arrest, which is particularly striking for longer genes[23,44,47]. We here show that the recovery of those transcriptionally arrested genes following repair by TCR requires the CSB-PAF1C axis for efficient recovery of productive elongation.

Our findings suggest a dual role for CSB. Firstly, CSB is an essential DNA repair factor in TCR that associates with DNA damage-stalled RNAPII and subsequently facilitates the recruitment of downstream TCR factors to initiate repair[14,15]. Secondly, CSB regulates transcription recovery from promoter-proximal sites, which involves the CSB-mediated association of PAF1C with RNAPII (Figs. 1 and 2). Knockdown of CDK9 was previously shown to not impair transcription recovery at late timepoints after UV[49], in line with the notion that is process is driven in a p-TEFb-independent manner. We show that PAF1C is dispensable for the repair of transcription-blocking DNA lesions, suggesting that the PAF1C-CSB interaction plays a unique role in transcription recovery (Figs. 3 and 4).

Both PAF1 and RNAPII shift into the first ~2 kb of gene bodies at 8 h after UV irradiation (Figs. 5 and 6). These changes could be

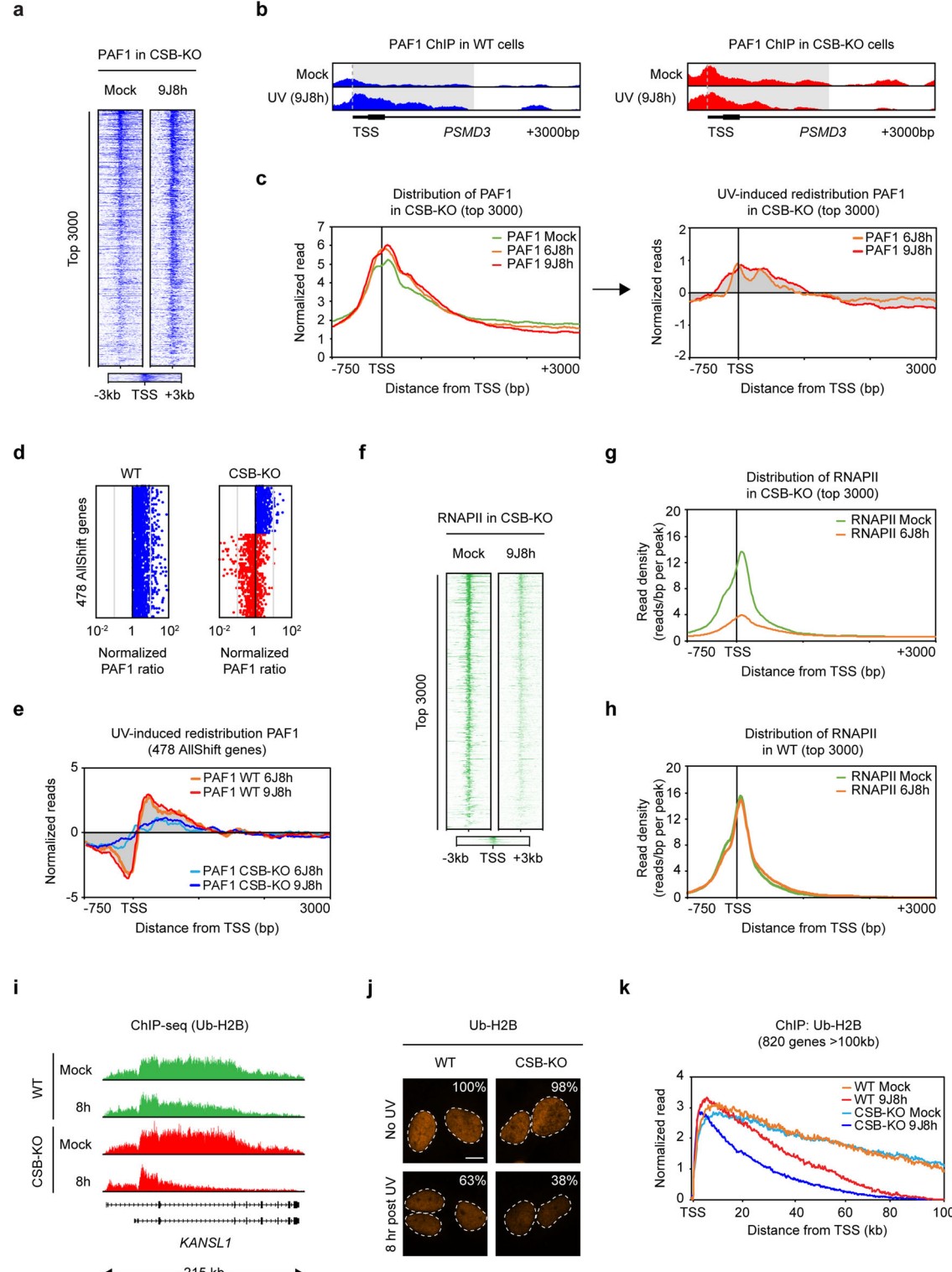

triggered by the strong UV-induced interaction between PAF1C and RNAPII, which is facilitated by CSB. In agreement with this model, we find that CSB-deficient cells fail to reposition PAF1 after UV. We propose that these changes reflect increased elongation from promoter-proximal sites, resulting in an increased number of RNAPII molecules within this region (Figs. 5 and 6). Consistent with this model, we find that genes that do not show a shift in PAF1 binding also do not release RNAPII, and that depletion of PAF1 prevents the UV-induced release of RNAPII from TSS sites (Fig. 6). Together, our findings suggest the

following model (Fig. 9): (I) CSB recruits PAF1C after UV and loads this complex onto RNAPII paused at TSS sites, (II) CSB recruits CSA to TSS sites after UV irradiation resulting in the ubiquitylation of ATF3, which is also bound near the TSS sites to repress transcription[11], (III) the activation of RNAPII by PAF1C promotes pause release and elongation activation that drives transcription recovery. Our findings suggest that PAF1 that is loaded onto RNAPII by CSB at promoter-proximal sites travels with RNAPII and facilitates efficient and productive elongation throughout the gene. Supporting this notion, we find that PAF1 at

**Fig. 7 The UV-induced repositioning of PAF1 and associated ubiquitylation of H2B requires CSB. a** Representative heatmaps around TSS from single PAF1 ChIP-seq data of the top 3000 genes that bind PAF1 in CSB-KO cells at 0 and 8 h after 9 J/m² UV. **b** UCSC genome browser track showing the read density of PAF1 signal across the *PSMD3* gene. Tracks represent pooled reads of three wild-type mock and three wild-type 9J8h PAF1 ChIP-seq replicates (blue) and two CSB-KO mock and two CSB-KO 9J8h PAF1 ChIP-seq replicates (red). **c** The left panel shows averaged metaplots of PAF1 ChIP-seq of the top 3000 genes in CSB-KO unirradiated cells (mock, $n = 2$) or CSB-KO cells 8h after UV irradiation with 6 J/m² ($n = 1$) and 9 J/m² ($n = 2$). The right panel shows the UV-induced redistribution of PAF1 calculated by subtracting the mock from the +UV distribution profiles. **d** Quantification of UV-induced PAF1 traveling ratios (or shift) in the 478 all shift genes as defined in WT cells in Fig. 6a, b. Genes shifting in all CSB-KO replicates are in blue, genes not shifting in at least 1 of the replicates are indicated in red. **e** The UV-induced redistribution of PAF1 calculated by subtracting the mock from the +UV distribution profiles in WT and CSB-KO cells. **f** Representative heatmaps around the TSS from single ChIP-seq data on RNAPII of the top 3000 genes that bind PAF1. Heatmaps are show for CSB-KO cells at 0 and 8 h after 6 J/m². **g** Averaged non-normalized metaplots around the TSS of RNAPII ChIP-seq of the top 3000 genes in unirradiated (mock, $n = 2$) or UV-irradiated (8 h after 6 J/m², $n = 2$) CSB-KO cells showing differences in total RNAPII binding in different conditions. **h** As in g for unirradiated wild-type cells (mock, $n = 3$) or UV-irradiated wild-type cells, 8 h after 6 J/m² ($n = 3$). **i** UCSC genome browser tracks showing the read density of ubiquitylated H2B (Ub-H2B) signal across the *KANSL1* gene in wild-type and CSB-KO cells 0 or 8 h after 9 J/m². **j** Representative images of U2OS WT or CSB-KO stained for Ub-H2B at 0 or 8 h after 9 J/m². Scale bar indicates 10 μm. Boxplots of the quantification of these images are presented in Supplementary Fig. 6. **k** Averaged metaplots of Ub-H2B ChIP-seq of 820 genes of >100 kb in WT or CSB-KO cells at 0 or 8 h after 9 J/m² UV. Total reads per plot were normalized to total Ub-H2B levels quantified by microscopy as in **j** and Supplementary Fig. 6. Data are averages of two replicates per condition.

TTS sites is strongly reduced shortly after UV, but reappears at 26 h after UV irradiation (Supplementary Fig. 4e). At this time, we also detect that RNA synthesis is restored near the end of genes in wild-type cells, but not in PAF1-depleted cells (Fig. 8). The precise mechanisms by which PAF1C stimulates processive transcription remain to be elucidated, but these likely involve modulating chromatin structure through histone marks, such as H2B ubiquitylation[5], H3K79 methylation[50], and H3K4 methylation, which have been associated with PAF1C activity. In line with this possibility, we find that H2B ubiquitylation is strongly reduced toward the end of long genes in CSB-deficient cells after UV irradiation (Fig. 8). Additionally, both H3K79 methylation and H3K4 methylation have been linked to transcription restart after UV in mouse cells[27], and *C. elegans*[51]. Whether these histone marks are deposited in a PAF1-dependent manner remains to be addressed. An intriguing possibility is that PAF1C is needed to stimulate transcription through former repair sites, which may have a chromatin signature that is suboptimal for transcription, such as H2A ubiquitylation[52].

## Methods

**Cell lines**. All cell lines are listed in Supplementary Table 1. All human RPE1-hTERT-Flp-In/T-Rex (RPE-hTERT(FRT)), U2OS, U2OS-Flp-In/T-Rex (U2OS (FRT)) and CSB-deficient CS1AN-SV cells were cultured at 37 °C in an atmosphere of 5% CO₂ in DMEM, supplemented with antibiotics, 10% fetal calf serum and glutaMAX (Gibco). Primary XP-C patient fibroblasts XP168LV were cultured at 37 °C in an atmosphere of 5% CO₂ in Ham's F10 medium without thymidine (Lonza) supplemented with 20% fetal calf serum and antibiotics.

Flp-In/T-REx cells (either RPE1-hTERT(FRT) or U2OS(FRT)) were used to stably express inducible version of GFP-tagged proteins by co-transfecting pCDNA5/FRT/TO-Puro plasmid encoding GFP-tagged fusion proteins (5 μg), together with pOG44 plasmid encoding the Flp recombinase (0.5 μg). After selection on 1 μg/mL puromycin, single clones were isolated and expanded. RPE1-hTERT-Flp-In/T-Rex were generated expressing either GFP-NLS or GFP-LEO1. U2OS-Flp-In/T-Rex (knockout for specific TCR genes; see below) were generated stably expressing CSA-GFP, GFP-CSB, UVSSA-GFP, or GFP-XPA in the corresponding KO line. In addition, U2OS-Flp-In/T-Rex were generated expressing siRNA-resistant WT PAF1 or PAF1 lacking seven amino acids required for its association with LEO1 (PAF1^ΔLEO1)[37]. Stable U2OS-Flp-In/T-REx or RPE1-hTERT-Flp-In/T-REx clones were incubated with 2 μg/mL doxycycline to induce expression of GFP-tagged proteins.

To generate cells sensitive to auxin-inducible degradation of PAF1, U2OS cells expressing TIR1 under the control of doxycycline (U2OS-TetOn-TIR1) were transfected with plasmids encoding Cas9 and an sgRNA close to the stop codon of the PAF1 gene, together with a donor plasmid (adjusted from pMK286, Addgene) containing an auxin-inducible degron (AID) and G418 cassette (AID-P2A-G418) flanked by ~1 kb arms homologous to the PAF1 locus (Supplementary Fig. 3a). This generated endogenously tagged U2OS-TetOn-TIR1-PAF-AID cells. Cells were selected with 200 μg/mL G418 for ~14 days and individual clones were selected and tested for auxin-inducible PAF1 degradation using western blot analysis. To induce depletion of PAF1, cells were induced to express TIR1 by ~24 h treatment with

2 μg/mL Doxycycline, followed by treatment with 500 μM auxin (3-Indoleacetic acid; Sigma) for 5–6 h.

**Generation of TCR knockout cells**. U2OS-Flp-In/T-Rex were co-transfected with pU6-gRNA:PGK-puro-2A-tagBFP (Sigma–Aldrich library from the LUMC) containing specific sgRNA targeting *CSB*, *CSA*, *UVSSA*, or *XPA* (Supplementary Table 2), together with pX458 (addgene) encoding Cas9. Cells were selected with puromycin (1 μg/ml) for 3 days and seeded at low density without puromycin. Individual clones were isolated and screened for loss of protein-of-interest expression and absence of stable Cas9 expression by western blot analysis and / or sanger sequencing.

**Plasmids**. pcDNA5/FRT/TO-Puro was purchased from Addgene. PCR was used to generate the following GFP fusion proteins, which were inserted into pcDNA5/FRT/TO-Puro: GFP-NLS, GFP-CSB, CSA-GFP, UVSSA-GFP, GFP-XPA, GFP-LEO1, GFP-CTR9, GFP-PAF1. Overlap PCR was used to generate GFP-PAF1 that was resistant to siPAF1-2 and siPAF1-3 by introducing the following silent mutations: 5-AAA CAA CAA TTC ACA GAA GAG-3 and 5-GAC GAC GTC TAC GAT TAT-3. Subsequently, this construct used to generate a GFP-PAF1 lacking five amino acids (202–207) required for its association with LEO1 (PAF1^ΔLEO1). pMK286 was purchased from Addgene. An extended multiple cloning site was introduced surrounding the AID-P2A-G418 using PCR. Around 1 kb of flanking sequences homologous to the PAF1 locus were introduced into this adjusted AID-P2A-G418 plasmid. All relevant primers are listed in Supplementary Table 3.

**Transfections**. Cells were transfected with plasmid DNA using Lipofectamine 2000 according to the manufacturer's instructions. Cells were typically imaged 24 h after transfection. All siRNA transfections (see list of siRNA sequences in Supplementary Table 4) were performed with 40 nM siRNA duplexes using Lipofectamine RNAiMAX (Invitrogen). Cells were transfected twice with siRNAs at 0 and 36 h and were typically analyzed 60 h after the first transfection.

**Western blotting**. Cell extracts were generated by cell lysis and boiled in sample buffer. Proteins were separated by sodium dodecyl sulfate polyacrylamide gel electrophoresis (SDS-PAGE) and transferred to PVDF membranes (EMD Millipore). Protein expression was analyzed by immunoblotting with the indicated primary antibodies (Supplementary Table 5) and secondary CF680 Goat Anti-Rabbit IgG antibody at 1:10,000 or CF770 Goat Anti-Mouse IgG antibody at 1:10,000, followed by detection using the Odyssey infrared imaging scanning system (LI-COR biosciences, Lincoln, Nebraska USA). Uncropped western blot figures are available in the source datafile provided with this paper.

**Clonogenic survival assays**. Cells were plated in low density in culture dishes, allowed to attach and treated with Illudin S at different concentrations for 72 h. Illudin S was removed and cells were allowed to form clones for 7–10 days. To visualize clones, cells were subjected to NaCl fixation and methylene blue staining. Cell survival after Illudin S treatment was defined as the percentage of cells able to form clones, relative to the untreated condition.

**Immunoprecipitation for Co-IP**. Except where indicated otherwise, all co-IP experiments were performed 1 h after UV irradiation. For endogenous RNAPII immunoprecipitation, cells were subjected to chromatin fractionation prior to immunoprecipitation. Cells were lysed in EBC-150 buffer (50 mM Tris, pH 7.5, 150 mM NaCl, 0.5% NP-40, 2 mM MgCl₂ supplemented with protease and

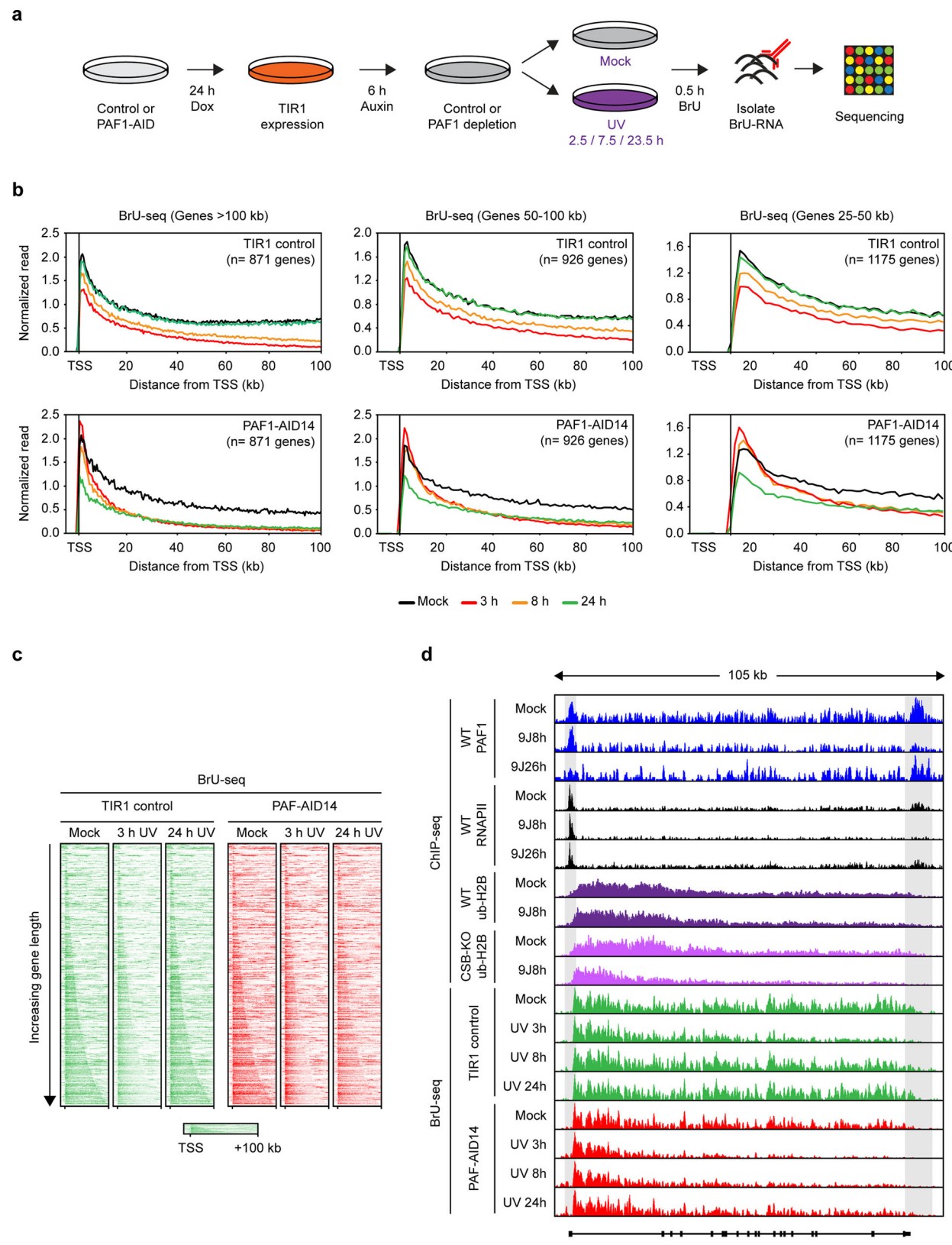

phosphatase inhibitor cocktails (Roche)) for 20 min at 4 °C, followed by centrifugation to remove cytoplasmic proteins. Subsequently, the chromatin fraction was solubilized in EBC-150 buffer with 500 U/mL Benzonase (Novagen) and 2 ug of antibody recognizing elongating RNAPII-S2 (Abcam; ab5095) or RNAPII-S5 (Abcam; ab5408) for 1 h at 4 °C under rotation. Next, the NaCl concentration of the lysis buffer was increased to 300 mM by adding concentrated (5 M) NaCl, and lysates were incubated for another 30 min at 4 °C. The lysates were cleared from

insoluble chromatin and were subjected to immunoprecipitation with protein A agarose beads (Millipore) for 1.5 h at 4 °C. The beads were then washed 4–6 times with EBC-300 buffer (50 mM Tris, pH 7.5, 300 mM NaCl, 0.5% NP-40, 1 mM EDTA) and boiled in sample buffer. Bound proteins were resolved by SDS-PAGE and immunoblotted with the indicated antibodies.

Immunoprecipitation of GFP-tagged proteins was performed using a similar protocol with the following exceptions: Cell pellets were directly solubilized in

**Fig. 8 PAF1C activates RNAPII pause release and transcription elongation after UV irradiation. a** Outline of the BrU-seq approach to measure nascent transcription across the genome. **b** Metaplots of nascent transcription in genes of >100 kb, between 50 and 100 kb, or between 25 and 50 kb in one replicate of either TIR1 cells (upper panels) or PAF1-AID cells (lower panels) that were either mock-treated, or UV-irradiated (7 J/m²) and analyzed at the indicated timepoints (3, 8, or 24 h). The relative distribution of nascent transcript read density (in reads per thousand base-pairs per million reads) was normalized to the absolute nascent transcript intensities measured in parallel to the BrU-seq experiments using the same cells and timepoints (see Fig. 4g, h). A replicate experiment is shown in Supplementary Fig. 7c. **c** Heatmaps of BrU-seq data from the first replicate of unirradiated (mock) or UV-irradiated (3 or 24 h after 7 J/m²) TIR1 control or PAF1-AID cells. Data was mapped and processed as for ChIP-seq and data is presented for the top 3000 genes with PAF1 binding at the TSS followed by ranking according to gene length. **d** UCSC genome browser track showing the nascent transcript read density across the *ZFR* gene in unirradiated and UV-irradiated TIR1 and PAF1-AID cells. Also shown are the PAF1, RNAPII, and Ub-H2B read densities for the same gene for comparison.

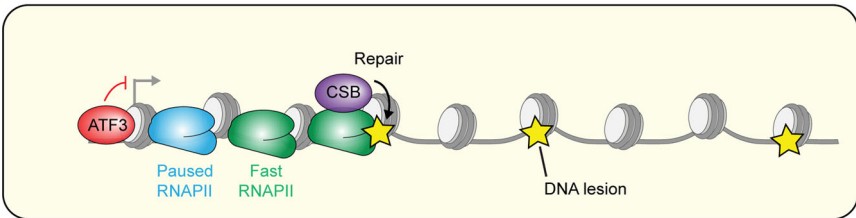

1 - 3 h after UV

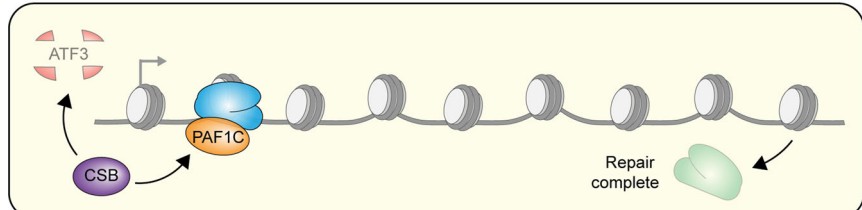

3 - 8 h after UV

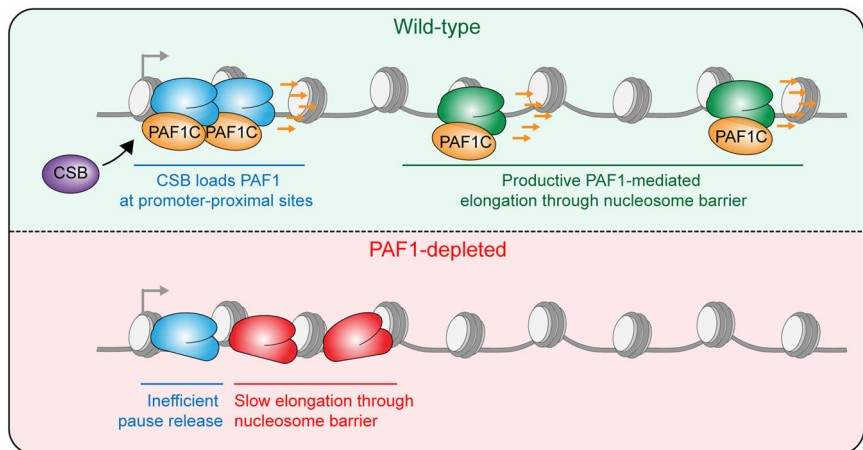

8 - 24 h after UV

**Fig. 9 Model of the role of PAF1C in restoring transcription across the genome after genotoxic stress.** (Top panel) Within the first 3 h after UV irradiation, RNAPII molecules stall at sites of DNA damages, inducing DNA repair through CSB. At the same time, the expression of transcriptional repressor ATF3 is strongly upregulated resulting in decreased transcription initiation at promoters. (Middle panel) CSB recruits the PAF1C to RNAPII paused around TSS sites and stimulates CSA-mediated degradation of ATF3. (Bottom panel) PAF1C binding to RNAPII around the TSS pause site subsequently promotes RNAPII pause release and stimulates productive elongation throughout genes.

EBC-150 supplemented with 500 U Benzonase, without chromatin fractionation, and pull-down was not dependent on antibody-mediated pull-down, but was performed using GFP Trap beads (Chromotek).

**Generation of mass spectrometry samples**. For stable isotope labeling by amino acids in cell culture (SILAC), CS1AN-SV expressing GFP-CSB were cultured for 14 days in media containing heavy (H) and light (L) labeled forms of the amino

acids arginine and lysine respectively. SILAC-labeled cells were mock-treated (L) or exposed to 20 J/m² UV-C light and allowed to recover for 1 h (H). Label-free mass spectrometry samples were also either kept untreated or exposed to 20 J/m² UV-C light and allowed to recover for 1 h. A pool of equal amounts of H- and L-labeled cells (SILAC) or individual label-free mass spectrometry samples, were subsequently subjected to immunoprecipitation using GFP Trap beads as described above. After pull-down, the beads were washed two times with EBC-300 buffer and two times with 50 mM (NH₄)₂CO₃ followed by overnight digestion using 2.5 μg

trypsin at 37 °C under constant shaking. Peptides of the H and L precipitates were mixed in a 1:1 ratio and all samples were desalted using a Sep-Pak tC18 cartridge by washing with 0.1% acetic acid. Finally, peptides were eluted with 0.1% formic acid/60% acetonitrile and lyophilized according to[53].

**Mass spectrometry data acquisition.** Mass spectrometry was performed essentially as previously described[54]. Samples were analyzed on a Q-Exactive Orbitrap mass spectrometer (Thermo Fisher, Germany) coupled to an EASY-nanoLC 1000 system (Proxeon, Odense, Denmark). For the SILAC samples, digested peptides were separated using a 13 cm fused silica capillary (ID: 75 μm, OD: 375 μm, Polymicro Technologies, California, US) in-house packed with 1.8 μm C18 beads (Reprospher-DE, Pur, Dr. Maisch, Ammerburch-Entringen, Germany). Peptides were separated by liquid chromatography using a gradient from 2 to 95% acetonitrile with 0.1% formic acid at a flow rate of 200 nl/min for 2 h. The mass spectrometer was operated in positive-ion mode at 1.8 kV with the capillary heated to 250 °C. Data-dependent acquisition mode was used to automatically switch between full scan MS and MS/MS scans, employing a top 10 method. Full scan MS spectra were obtained with a resolution of 70,000, a target value of $3 \times 10^6$ and a scan range from 400 to 2000 $m/z$. Higher-Collisional Dissociation (HCD) tandem mass spectra (MS/MS) were recorded with a resolution of 17,500, a target value of $1 \times 10^5$ and a normalized collision energy of 25%. Maximum injection times for MS and MS/MS were 20 and 60 ms, respectively. For label-free samples, digested peptides were separated using a 15 cm fused silica capillary (ID: 75 μm, OD: 375 μm, Polymicro Technologies, California, US) in-house packed with 1.9 μm C18-AQ beads (Reprospher-DE, Pur, Dr. Maisch, Ammerburch-Entringen, Germany). Peptides were separated by liquid chromatography using a gradient from 2 to 95% acetonitrile with 0.1% formic acid at a flow rate of 200 nl/min for 90 min. The mass spectrometer was operated in positive-ion mode at 2.8 kV with the capillary heated to 250 °C. Data-dependent acquisition mode was used to automatically switch between full scan MS and MS/MS scans, employing a top seven method. Full scan MS spectra were obtained with a resolution of 70,000, a target value of $3 \times 10^6$ and a scan range from 400 to 2000 $m/z$. Higher-Collisional Dissociation (HCD) tandem mass spectra (MS/MS) were recorded with a resolution of 35,000, a target value of $1 \times 10^5$ and a normalized collision energy of 25%. Maximum injection times for MS and MS/MS were 50 and 120 ms, respectively. For all samples, the precursor ion masses selected for MS/MS analysis were subsequently dynamically excluded from MS/MS analysis for 60 sec. Precursor ions with a charge state of 1 or greater than 6 were excluded from triggering MS/MS events.

**Mass spectrometry data analysis.** Raw mass spectrometry data were further analysed in MaxQuant v 1.5.3.30 according to Tyanova et al.[55] using standard settings with the following modifications. For the SILAC-labeled GFP-CSB samples, multiplicity was set to 2, marking Arg10 and Lys8 as heavy labels. Maximum missed cleavages by trypsin was set to 4. Searches were performed against an in silico digested database from the human proteome including isoforms and canonical proteins (Uniprot, 18 June 2018). Minimum peptide length was set to 6 aa and maximum peptide mass was set to 5 kDa. Carbamidomethyl (C) was disabled as fixed modification. The match between runs feature was activated. Minimum ratio count for quantification was set to 1. For the label-free GFP-LEO1 and GFP-RBP1 samples, maximum missed cleavages by trypsin was set to 4. Label-free quantification was activated, not enabling Fast LFQ. Searches were performed against an in silico digested database from the human proteome including isoforms and canonical proteins (Uniprot, 18 June 2018). Carbamidomethyl (C) was disabled as fixed modification. The match between runs feature was activated and iBAQ quantification was also enabled. MaxQuant output data from the SILAC samples analysis was further processed in Microsoft Excel 2016 for comprehensive visualization. Label-free analysis was further carried out in the Perseus Computational Platform v1.5.5.3 according to Tyanova et al.[56]. LFQ intensity values were log2 transformed and potential contaminants and proteins identified by site only or reverse peptide were removed. Samples were grouped in experimental categories and proteins not identified in four out of four replicates in at least one group were also removed. Missing values were imputed using normally distributed values with a 1.8 downshift (log2) and a randomized 0.3 width (log2) considering whole matrix values. Two-sided $t$-tests were performed to compare groups. Analyzed data were exported from Perseus and further processed in Microsoft Excel 2016 for comprehensive visualization.

**Nascent transcript level measurements.** Cells were plated in DMEM supplemented with 10% Fetal Calf Serum (FCS) and, if needed, transfected with siRNAs as described above. Subsequently, cells were placed in DMEM supplemented with 1% FCS for at least 24 h prior to the nascent transcript measurement to reduce the excess of available uridine in the culture medium. Cells were UV irradiated, allowed to recover for the indicated time periods, and pulse-labeled with 400 μM 5-ethynyl-uridine (EU; Axxora) for 1 h. After medium-chase with DMEM without supplements for 15 min, cells were fixed with 3.7% formaldehyde in phosphate-buffered saline (PBS) for 15 min and stored in PBS. Nascent RNA was visualized by click-it chemistry, labeling the cells for 1 h with a mix of 60 μM atto azide-Alexa594 (Atto Tec), 4 mM copper sulfate (Sigma), 10 mM ascorbic acid (Sigma) and

0.1 μg/mL DAPI in a 50 mM Tris-buffer. Cells were washed extensively with PBS and mounted in Polymount (Brunschwig).

**Global-genome unscheduled DNA synthesis.** Cells were plated in DMEM supplemented with 10% Fetal Calf Serum (FCS) and, if needed, transfected with siRNAs as described above. Subsequently, cells were placed in DMEM supplemented with 1% FCS for at least 24 h prior to UV irradiation to reduce the excess of available deoxy-uridine in the culture medium. Cells were locally UV irradiated through 5 μm pore filters (Milipore; TMTP04700) with 30 J/m², and immediately pulse-labeled with 20 μM 5-ethynyl-deoxy-uridine (EdU; VWR) and 1 μM FuDR (Sigma–Aldrich) for 1 h. After medium-chase with DMEM containing 10 μM Thymidine for 30 min, cells were fixed with 3.7% formaldehyde in PBS for 15 min at room temperature and stored in PBS. Next, cells were permeabilized for 20 min in PBS with 0.5% Triton-X100 and blocked with with 3% BSA (Thermo Fisher) in PBS. The incorporated EdU was visualized by click-it chemistry, labeling the cells for 1 h with a mix of 60 μM atto azide-Alexa 647 (Atto Tec), 4 mM copper sulfate (Sigma) and 10 mM ascorbic acid (Sigma) in a 50 mM Tris-buffer. After this, the cells were post-fixed with 2% PFA for 10 min and blocked with 100 mM Glyine. Cells were washed extensively with PBS, DNA was denatured with 0.5 M NaOH for 5 min, blocked with 10% BSA (Thermo Fisher) in PBS for 15 min and equilibrated in 0.5%BSA and 0.05% TritonX100 in PBS (WB-buffer). Damaged areas were visualized by labeling the cells for 2 h with mouse anti-CPD (Cosmo Bio; CAC-NM-DND-001; 1:1000 in WB-buffer). After primary antibody incubation, cells were washed extensively with WB-buffer, stained with goat anti-rabbit IgG-Alexa 555 (Thermo Fisher; A-21424; 1:1,000 in WB-buffer) for 1 h, again washed extensively with WB-buffer, counterstained with 0.1 μg/mL DAPI, washed extensively with PBS and mounted in Polymount (Brunschwig).

**TCR-specific unscheduled DNA synthesis.** Detection of TCR-specific unscheduled DNA synthesis was performed essentially as previously described[39]. Primary XP168LV (XP-C patient cells) were transfected with siRNAs and subsequently serum starved for at least 24 h in F10 medium (Lonza) supplemented with 0.5% FCS and antibiotics. Cells with subsequently irradiated with 8 J/m² UV-C, and pulse-labeled with 20 μM 5-ethynyl deoxy-uridine (EdU; Invitrogen) and 1 μM FuDR (Sigma–Aldrich) for 8 h. After labeling, cells were chased with F10 medium supplemented with 0.5% FCS and 10 μM thymidine for 15 min, and fixed for 15 min with 3.6% formaldehyde and 0.5% Triton-X100. Next, cells were permeabilized for 20 min in PBS with 0.5% Triton-X100 and washed and stored in 3% bovine serum albumin (BSA, Thermo Fisher) in PBS. The incorporated EdU was visualized by click-it chemistry-mediated binding of Biotin (Azide-PEG3-Biotin Conjugate; Jena Biosciences) using the protocol and reagents from the Invitrogen Click-iT EdU Cell Proliferation Kit for Imaging (Invitrogen), and signals were amplified using protocol and reagent of the Alexa Fluor-488 Tyramide streptavidin SuperBoos Kit (Thermo Fisher). After click-it and amplification, cells were counterstained with 0.1 μg/mL DAPI, washed extensively with 0.1% Triton-X100 in PBS and mounted in Polymount (Brunschwig).

**Immunostaining of ubiquitylated H2B levels.** Cells were plated in DMEM supplemented with 10% Fetal Calf Serum (FCS). Subsequently, cells were UV irradiated with 9 J/m², or kept untreated. At 8 h after UV irradiation, cells were washed with cold PBS, fixed with 100% methanol on ice for 10 min, extensively washed and stored in PBS. From here all steps were performed at room temperature. Cells were further permeabilized by incubation with 0.5% TritonX100 in PBS for 5 min. Then nuclei were consecutively blocked with 100 mM Glyine in PBS for 10 min, washed extensively with PBS and blocked with 0.5% BSA and 0.05% tween20 in PBS (WB-buffer) for 10 min. Ub-H2B was visualized by labeling the cells for 2 h with rabbit anti-ub-H2B (K120) (Cell signaling (mAb#5546, D11); 1:200 in WB-buffer). After primary antibody incubation, cells were washed extensively with WB-buffer, stained with goat anti-rabbit IgG-Alexa 555 (Thermo Fisher; A-21429 1:1,000 in WB-buffer) for 1 h, again washed extensively with WB-buffer, counterstained with 0.1 μg/mL DAPI, washed extensively with PBS and mounted in Polymount (Brunschwig).

**Microscopic analysis of fixed cells.** Images of fixed samples were acquired on a Zeiss AxioImager M2 or D2 widefield fluorescence microscope equipped with ×63 PLAN APO (1.4 NA) oil-immersion objectives (Zeiss) and an HXP 120 metal-halide lamp used for excitation. Fluorescent probes were detected using the following filters: DAPI (excitation filter: 350/50 nm, dichroic mirror: 400 nm, emission filter: 460/50 nm), Alexa 555 (excitation filter: 545/25 nm, dichroic mirror: 565 nm, emission filter: 605/70 nm), Alexa 647 (excitation filter: 640/30 nm, dichroic mirror: 660 nm, emission filter: 690/50 nm). Images were recorded using ZEN 2012 (blue edition, version 1.1.0.0) software and analyzed in Image J (1.48 v).

**ChIP sequencing.** Cells were plated and grown to ~90% confluency and cross-linked with 0.5 mg/mL disuccinimidyl glutarate (DSG; Thermo Fisher) in PBS for 45 min at room temperature. Cells were washed with PBS and crosslinked with 1% PFA for 20 min at room temperature. Fixation was stopped by adding 1.25 M Glycin in PBS to a final concentration of 0.1 M for 3 min at room temperature. Cells were washed with cold PBS and lysed and collected in a buffer containing 0.25% Triton X-100, 10 mM EDTA (pH 8.0), 0.5 mM EGTA (pH 8.0), and 20 mM

Hepes (pH 7.6). Chromatin was pelleted in 5 min at $400 \times g$ and incubated in a buffer containing 150 mM NaCl, 1 mM EDTA (pH 8.0), 0.5 mM EGTA (pH 8.0) and 50 mM Hepes (pH 7.6) for 10 min at 4 °C. Chromatin was again pelleted for 5 min at $400 \times g$ and resuspended in ChIP-buffer (0.15 % SDS, 1 % Triton X-100, 150 mM NaCl, 1 mM EDTA (pH 8.0), 0.5 mM EGTA (pH 8.0) and 20 mM Hepes (pH 7.6)) to a final concentration of $15 \times 10^6$ cells/ml. Chromatin was sonicated to approximately one nucleosome using a Bioruptor waterbath sonicator (Diagenode). Chromatin of ~$5 \times 10^6$ cells was incubated with 3-ug antibody (RNAPII, rabbit polyclonal, Bethyl laboratories, A304-405A; PAF1, rabbit polyclonal, Bethyl laboratories, A300-172A; Ub-H2B(Lys120), rabbit monoclonal (D11), Cell signaling, #5546) overnight at 4 °C, followed by a 1.5 h protein-chromatin pull-down with a 1:1 mix of protein A and protein G Dynabeads (Thermo Fisher; 10001D and 10003D). ChIP samples were washed extensively, followed by decrosslinking for 4 h at 65 °C in the presence of proteinase K. DNA was purified using a Qiagen MinElute kit. Sample libraries were prepared using Hifi Kapa sample prep kit and A-T mediated ligation of Nextflex adapters or xGen UDI-UMI adapters. Samples were sequenced using an Illumina NextSeq500 or HiSeq X, using paired-end sequencing with 42 or 151 bp from each end.

**ChIP-seq analyses**. A sequencing quality profile was generated using FastQC (Version 0.11.2). Reads were aligned to the Human Genome 38 (Hg38; https://ftp.ncbi. nlm.nih.gov/genomes/all/GCA/000/001/405/GCA_000001405.15_GRCh38/seqs_for_ alignment_pipelines.ucsc_ids/GCA_000001405.15_GRCh38_no_alt_analysis_set.fna. gz) using bwa-mem tools (BWA (Version 0.7.16a))[57]. Only high-quality reads (> q30) were included in the analyses and duplicates were removed using Samtools (Version 1.6) with fixmate -m and markdup -r settings (Supplementary Table 6). Bedgraph UCSC genome tracks were generated and PAF1 binding peaks were identified using the callpeaks tool of MACS2 (Version 2.1)[58], correlating each ChIP-seq sample with its UV-dose-associated input, with standard tool-settings. Example genome tracks were generated in IGV (Version 2.4.3). Bam files were converted into TagDirectories using HOMER tools[59].

A list of 49,948 transcription start sites was obtained from the UCSC genome database (https://genome.ucsc.edu/cgi-bin/hgTables) selecting the knownCanonical table containing the canonical transcription start sites per gene. For PAF1 and RNAPII ChIP-seq, only genes of 3–100 kb were included in the analyses to prevent the inclusion of extremely small genes that might not be damaged under our experimental conditions, or genes that likely will acquire multiple DNA damages and might therefore not represent repaired genes in the timeframe of our experiments. To prevent contamination of binding profiles, genes should be nonoverlapping with at least 2 kb between genes. A total of 8811 genes were selected.

For Ub-H2B ChIP-seq 820 genes of >100 kb were selected that were also not overlapping with other genes with at least 2 kb between genes. Binding profiles within selected areas of individual genes (e.g., around TSS or TTS), were defined using the AnnotatePeaks.pl tool of HOMER using the default normalization to 10mln reads. Metagene profiles were defined using the makeMetaGeneProfile.pl tool of HOMER (Version 4.8.2), using default settings. Individual datasets were subsequently processed in R (Version 3.5.3) and Rstudio (Version 1.1.423)[60]. First, read densities in input samples were subtracted from individual ChIP-seq datasets to background-correct our data in which negative values were converted to 0, to prevent the use of impossible negative read densities in further calculations. ChIP profiles were averaged per sets of genes. PAF1 and RNAPII profiles were normalized to area under the curve to allow proper comparison of the profiles without effects of overall differences in read density. Traveling ratios were defined per gene over ranges indicated in individual analyses, with infinite ratios removed. Ub-H2B ChIP profiles were first background-subtracted (background at 200 bp before the TSS), and subsequently area under the curve of the plots were normalized to Ub-H2B levels defined by microscopy (as described above). Original ChIP-seq datafiles for RNAPII, ATF3, CSA and CSB, were also obtained from Epanchintsev and colleagues[11] (GSE87562) and Hou and colleagues[5] (GSE116169), and data for PAF1 was obtained from Chen and colleagues[41] (GSE97527) (GEO; https://www.ncbi.nlm.nih.gov/geo/). These files were subsequently converted into FASTQ files using NCBI sratoolkit.2.9.6-1-win64 and processed as described above, but without subtraction of Input reads.

**BrU-sequencing and data analysis**. U2OS TIR1 or PAF1-AID cells were induced with doxycycline for 24 hrs, and subsequently with auxin for 5 h. After this treatment, cells were either mock-treated or irradiated with UV-C light (7 J/m²). Cells were then incubated in conditioned media for different periods of time (0, 3, 8, and 24 h) before being incubated with 2 mM bromouridine (BrU) at 37 °C for a 30 min. The cells were then lysed in TRIzol reagent (Invitrogen) and BrU-containing RNA was isolated as previously described[23]. cDNA libraries were made from the BrU-labeled RNA using the Illumina TruSeq library kit and paired-end 151 bp sequenced using the Illumina NovaSeq platform at the University of Michigan DNA Sequencing Core. Data were processed as previously described[61]. Briefly, reads were prefiltered by alignment to the human ribosomal repeating subunit (GenBank U13369.1) and human mitochondrial genome (chrM) from the hg38 reference genome using bowtie2 (Version 2.3.3.1). The remaining reads were mapped to the hg38 reference genome using STAR (Version 2.7.0 f) (Supplementary Table 6). Base coverages were used to compute read counts for features, such as genes and bins, which were then normalized to feature length and number

of uniquely-mapped reads (RPKM method). Gene selection was based on the following criteria: TSS is at least 10 kb apart, expression is at least 0.05 RPKM and gene length of 25–50 kb ($n = 1{,}175$), 50–100 kb ($n = 926$), or at least 100 kb ($n = 871$). The median expression was calculated for each 500 bp bins from 5 kb upstream until 25, 50, or 100 kb downstream of the selected genes for each time-point. Average signal in $-5$ kb to TSS was put to 0. Aggregate plots were subsequently normalized to nascent transcript levels, as quantified by 5-EU labeling, relative to the control in their specific cell type. For example, PAF-AID cells 3 h after UV irradiation showed 62% RNA relative to PAF-AID control cells, so we multiplied the expression of the bins by 0.62. Heatmaps and UCSC tracks were generated by mapping and processing data as described for ChIP-seq analyses.

**Reporting summary**. Further information on research design is available in the Nature Research Reporting Summary linked to this article.

## Data availability
Mass spectrometry proteomics data are presented in main Fig. 1a, b, e, g, and Supplementary Figs. S1d, h and S2f, and have been deposited to the ProteomeXchange Consortium via the PRIDE partner repository with the dataset identifier PXD016198[62]. ChIP-seq data are presented in main Figs. 5–7 and Supplementary Figs. 4–7a, b. BrU-seq data are presented in mains Fig. 8b–d and Supplementary Figs. 7c, d. Both raw and processed ChIP-seq and BrU-seq data are deposited in the Gene Expression Omnibus under GSE140930. Additionally, previously published, publicly available, ChIP-seq datasets for RNAPII, ATF3, CSA, CSB, and PAF1 (https://www.ncbi.nlm.nih.gov/geo/, GSE87562, GSE116169, GSE97527), and previously published protein structure data (https://www.rcsb.org/, 5VVR, 6GMH) have been obtained and used in this manuscript, as well as reference datasets of the Hg38 genome (https://ftp.ncbi.nlm.nih.gov/genomes/ all/GCA/000/001/405/GCA_000001405.15_GRCh38/seqs_for_alignment_pipelines. ucsc_ids/GCA_000001405.15_GRCh38_no_alt_analysis_set.fna.gz), human ribosomal repeating subunit (GenBank U13369.1), and human mitochondrial genome (chrM) and the knownCanonical gene table from the UCSC genome database (https://genome.ucsc. edu/cgi-bin/hgTables, hg38 genome). Additional data will be made available upon reasonable request. Source data are provided with this paper.

## Code availability
Custom code will be made available upon reasonable request.

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

## Acknowledgements

We acknowledge Haico van Attikum for providing U2OS GFP-RPB1[63] and U2OS TIR1 cells, Leon Mullenders for providing CS1AN-SV40 cells expressing GFP or GFP-CSB, Rick Wood for his generous gift of XPA antibody, and Brian Magnuson for valuable help with the BrU-seq analysis. This work was funded by an LUMC Research Fellowship and an NWO-VIDI grant (ALW.016.161.320) to M.S.L., a Leiden University Fund (LUF) grant to DvdH (W18355-2-EM), an NWO-VENI grant to C.G.S., an ERC grant to A.C.O.V. (310913), and a Dutch Cancer Society (KWF-Young Investigator Grant: 11367) to R.G.-P. The MV and JAM labs are part of the Oncode Institute, which is partly funded by the Dutch Cancer Society (KWF).

## Author contributions

D.v.d.H. generated knockout cells, constructs and stable cell lines, PAF1-AID knockin cells, performed clonogenic survivals, PCR and western blot analysis to validate knockouts, RRS experiments, immunofluorescence on Ub-H2B, Co-IP experiments for western blot analysis, Co-IP experiments for mass spectrometry, ChIP experiments and analysis, BrU pulse-labeling experiments and analysis, and wrote the paper. C.G.S. performed ChIP-seq experiments, and measured and analyzed MS experiments in Fig. 1B. M.V. supervised C.G.S. and provided the infrastructure for ChIP-seq experiments. R.G.-P. and A.C.O.V. analyzed all MS experiments except those in Fig. 1b. A.K. and Y.v.d.W. generated knockout cells, stable cell lines, and performed RRS experiments. K.A. generated knockout cells and stable cell lines, and performed UDS experiments. M.D. performed SILAC-based MS. M.T.P. captured nascent RNA and performed library preparations for the BrU-seq experiments. K.Y. and M.L. analyzed the BrU-seq data. D.Z.

and J.A.F.M. performed and analyzed TCR-UDS experiments. H.W. and L.D. provided tools for ChIP-seq analyses. M.S.L. conceived and supervised the project and wrote the paper.

## Competing interests

The authors declare no competing interests.
