## [Peer Review File · Nature Communications]

REVIEWER COMMENTS

Reviewer #1 (Remarks to the Author):

The authors claim that recovery of RNA synthesis in UV damaged cells, which is a known defect associated with CSA and CSB mutated cells, occurs independently of the TCR defect that is also known to be associated with these cells. They conduct experiments which suggest that recovery of RNA synthesis in response to UV damage is dependent upon a UV-induced interaction between PAF1C and the CSB protein. They also show that disruption of this interaction in PAF1C depleted cells does not affect the repair of UV damage. Finally, they conduct ChIP-seq experiments that appear to support a model in which PAF1C stimulates the release of paused RNAPolIII in promoter-proximal regions (there is no evidence of how this might occur), and subsequently as a processivity factor, which promotes transcription elongation throughout genes. This is summarised in a final model described in the manuscript. If the interpretation of the results is correct, then the findings provide a significant advance in our understanding of the recovery of RNA synthesis in response to DNA damage in cells. To confirm that their interpretation of the results is correct, however, the authors should determine whether failure to recover RNA synthesis occurs in PAF1C depleted cells, where a level of repair remains active. For example, in CSA cells, where importantly they show the CSB/PAF1C interaction remains intact, TCR is defective - but removal of UV lesions will occur by the functional GGR pathway. Failure to recover RNA synthesis in UV-treated PAF1C depleted CSA cells would confirm their interpretation of the current results, and remove any concerns over alternative explanations of the existing data.

Reviewer #2 (Remarks to the Author):

In the present study, van den Heuvel et al. propose a role of CSB-mediated PAF1C recruitment in facilitating transcriptional restart following UV-induced TBLs. In support of their model, the authors show UV-induced interaction between PAF1 complex and RNAPII subunit Rpb1 in series of well-controlled SILAC-based and immunoprecipitation/Western blotting experiments. This interaction is dependent on TC-NER protein CSB, but not other repair proteins. PAF1 downregulation by either RNAi or AID system clearly demonstrated prolonged transcriptional arrest, while accompanying ChIP-seq experiments suggested PAF1-mediated repositioning of RNAPII into the promoter-proximal region by 3 and 8 hours post the UV treatment.

The present study provides interesting insights how PAF1 promotes transcriptional restart in response to UV stress and will be of interest for the researchers focusing on DNA damage response as well as regulation of transcription. The manuscript is well written and results and figures of very high quality.

I have some suggestions to further improve the manuscripts before its publication in Nature Communications:

1. I appreciate the concise way in which the manuscript was written but at some parts in the introduction and discussion, additional description of current literature about the regulation of transcription in response to TBLs and involved complexes will help the reader to understand the novelty of the present study. In line with this, the authors should better explain the proposed model in the discussion as well as in the figure legend and put the findings in the context of current literature.
2. The authors should address the effects of PAF1 depletion and/or overexpression on CSB-mediated ubiquitylation and stability for both Rpb1 pSer 2 and pSer 5.
3. Can authors take advantage of the existing UV/GFP-Rpb1 interactome in the presence and absence of CSB KO (Fig S2.F) to identify additional regulators of PAF1-Rpb1 interactions? Does CSB facilitate PAF1-Rpb1 interaction via ubiquitylation? The authors should at least speculate about potential regulators. The result in figure 2B should be repeated and shown on the same gel. This is an important piece of data to show that the interaction is not regulated by CDK9.
4. Additionally, authors also need to address the effects of CSB on PAF1 localization relative to Rpb1 in the genome and to correlate such observation to the established PAF1-mediated H2B ubiquitylation in promoting RNAPII processivity (Wu et al, 2014). Do the loss of CSB impair PAF1 recruitment to the genome and hence, PAF1-mediated H2B ubiquitylation?
5. To authors should strengthen the assessment for the impact of PAF1 complex on the repair of

CPD and 6,4-PP, as well as general cellular viability in addition to the TCR-UDS by using the AID system.

6. To authors should examine the impact of PAF1 on the genome localization and repositioning of RNAPII pSer2 and, in particular, pSer5 post UV. Metaplot data in Fig S4.D showed reduction RNAPII upstream of TSS in PAF-AID14 in addition to the loss of RNAPII in the proximal-promoter region. Recent study however reported that reduction of RNAPII elongation speed due to PAF1 depletion caused accumulation of pSer5 in the first 20 to 30 kb in the gene body (Hou et al, 2019).

Reference

Hou L, Wang Y, Liu Y, Zhang N, Shamovsky I, Nudler E, Tian B & Dynlacht BD (2019) Paf1C regulates RNA polymerase II progression by modulating elongation rate. *Proceedings of the National Academy of Sciences of the United States of America* 116: 14583–14592

Wu L, Li L, Zhou B, Qin Z & Dou Y (2014) H2B Ubiquitylation Promotes RNA Pol II Processivity via PAF1 and pTEFb. *Molecular Cell* 54: 920–931

REVIEWER COMMENTS

Reviewer #1 (Remarks to the Author):

The authors claim that recovery of RNA synthesis in UV damaged cells, which is a known defect associated with CSA and CSB mutated cells, occurs independently of the TCR defect that is also known to be associated with these cells. They conduct experiments which suggest that recovery of RNA synthesis in response to UV damage is dependent upon a UV-induced interaction between PAF1C and the CSB protein. They also show that disruption of this interaction in PAF1C depleted cells does not affect the repair of UV damage. Finally, they conduct ChIP-seq experiments that appear to support a model in which PAF1C stimulates the release of paused RNAPolIII in promoter-proximal regions (there is no evidence of how this might occur), and subsequently as a processivity factor, which promotes transcription elongation throughout genes. This is summarised in a final model described in the manuscript. If the interpretation of the results is correct, then the findings provide a significant advance in our understanding of the recovery of RNA synthesis in response to DNA damage in cells.

To confirm that their interpretation of the results is correct, however, the authors should determine whether failure to recover RNA synthesis occurs in PAF1C depleted cells, where a level of repair remains active. For example, in CSA cells, where importantly they show the CSB/PAF1C interaction remains intact, TCR is defective - but removal of UV lesions will occur by the functional GGR pathway. Failure to recover RNA synthesis in UV-treated PAF1C depleted CSA cells would confirm their interpretation of the current results, and remove any concerns over alternative explanations of the existing data.

We would like to point out that recovery of RNA synthesis (RRS) after UV irradiation is fully dependent on transcription-coupled repair (TCR). The global genome repair (GGR) machinery that is still active in TCR-deficient cells does not contribute in any way to transcription recovery. To demonstrate this, we have included RRS experiments in CSA knockout cells, which are deficient in TCR, but fully proficient in GGR. As shown in Supplementary Figure S3d (and below), CSA-KO cells are fully impaired in transcription recovery.

To address potential epistasis between the TCR pathway and the PAF1C pathway, we use control cells (TIR1) and PAF1-AID knockin cells to conditionally deplete endogenous PAF1, and additionally transfected cells with siRNAs against XPA. While control TIR cells transfected with control siRNAs against Luciferase (siLuc) show normal transcription recovery, PAF1-depleted cells are unable to recover transcription. When transfected with siRNAs targeting XPA, both TIR1 control cells and PAF1-depleted cells show impaired transcription recovery to the same extent (Supplementary Figure S3d; and below). We show in Figure 2d that the PAF1-RNAPII interaction still takes place in CSA-KO as well as XPA-KO cells. We conclude from this that (*page 8, line 27- 32*):

“knock-down of XPA in either TIR1 control cells or PAF1-depleted cells impaired transcription recovery to the same extent (Supplementary Figure 3d), suggesting that TCR and PAF1-mediated transcription restart are in the same pathway. Considering that PAF1 is not directly involved in TCR (Figure 3f), our findings suggest that PAF1-mediated transcription restart occurs after the elimination of transcription-blocking DNA lesions by TCR.”

Nascent transcript (EU) labeling

Reviewer #2 (Remarks to the Author):

In the present study, van den Heuvel et al. propose a role of CSB-mediated PAF1C recruitment in facilitating transcriptional restart following UV-induced TBLs. In support of their model, the authors show UV-induced interaction between PAF1 complex and RNAPII subunit Rpb1 in series of well-controlled SILAC-based and immunoprecipitation/Western blotting experiments. This interaction is dependent on TC-NER protein CSB, but not other repair proteins. PAF1 downregulation by either RNAi or AID system clearly demonstrated prolonged transcriptional arrest, while accompanying ChIP-seq experiments suggested PAF1-mediated repositioning of RNAPII into the promoter-proximal region by 3 and 8 hours post the UV treatment.

The present study provides interesting insights how PAF1 promotes transcriptional restart in response to UV stress and will be of interest for the researchers focusing on DNA damage response as well as regulation of transcription. The manuscript is well written and results and figures of very high quality.

I have some suggestions to further improve the manuscripts before its publication in Nature Communications:

1. I appreciate the concise way in which the manuscript was written but at some parts in the introduction and discussion, additional description of current literature about the regulation of transcription in response to TBLs and involved complexes will help the reader to understand the novelty of the present study. In line with this, the authors should better explain the proposed model in the discussion as well as in the figure legend and put the findings in the context of current literature.

We added additional descriptions of the current literature in the introduction and the discussion, and we provided additional information throughout the revised manuscript (marked in red).

2. The authors should address the effects of PAF1 depletion and/or overexpression on CSB-mediated ubiquitylation and stability for both Rpb1 pSer 2 and pSer 5.

We assessed the association of CSB and the ubiquitylation of the RPB1 subunit of RNAPII in PAF1-depleted cells in time-course experiments. Both CSB binding and RPB1 ubiquitylation still occurred in PAF1-depleted cells, albeit slightly reduced (Figure 3c; below). We conclude that (page 7, line 21-27):

“Pull down experiments of RNAPII-S2 at multiple time-points after UV irradiation revealed that CSB still interacted with RNAPII after UV, and that RPB1 ubiquitylation could also be detected in PAF1-depleted cells with kinetics similar as in TIR1 control cells (Figure 3c). The amount of CSB and ubiquitylated RNAPII were slightly reduced in PAF1-depleted cells, suggesting that PAF1C is not essential for, but may stabilize the interaction between CSB and RNAPII after UV (Figure 3c).”

The levels of RPB1-Ser2 or RPB1-Ser5 were not altered in PAF1-depleted cells compared to TIR1 control cells (Supplementary Figure S3b).

3. Can authors take advantage of the existing UV/GFP-Rpb1 interactome in the presence and absence of CSB KO (Fig S2.F) to identify additional regulators of PAF1-Rpb1 interactions? Does CSB facilitate PAF1-Rpb1 interaction via ubiquitylation? The authors should at least speculate about potential regulators.

Our GFP-RPB1 mass spectrometry datasets in WT and CSB-KO cells unfortunately did not reveal clear CSB-dependent and UV-induced interactors of RPB1 other than the PAF1 complex. All interactome data is deposited in the PRIDE repository for readers to re-analyse and browse.

It seems very unlikely that PAF1C association is mediated by CSB-dependent RPB1 ubiquitylation. We have recently shown that CSA is an important regulator of this modification (Nakazawa *et al.*, 2020, *Cell*), but CSA-KO cells, which are largely impaired in RPB1 ubiquitylation, show a normal PAF1-RNAPII interaction (Figure 2d).

To try and better understand how CSB facilitates the PAF1 binding, we compared available cryo-EM structures of yeast RAD26 (the orthologue of CSB) and the human PAF1C complex bound to RNAPII (Figure 3a). These structures are compatible with an interaction between PAF1C subunit LEO1 and CSB. To test this possibility, we generated stable cells expressing a mutant of PAF1 that is unable to interact with LEO1. We indeed find that this mutant of PAF1 has a strongly decreased interaction with CSB and RNAPII after UV (Figure 3b; see below). We conclude in the revised manuscript that (page 7, lines 11-13):

“PAF1-ΔLEO1 interacted much less efficiently with CSB, CSA, and RNAPII after UV irradiation compared to PAF1-WT, suggesting that the LEO1 subunit likely anchors the PAF1C complex to CSB and RNAPII after UV.”

4. The result in figure 2B should be repeated and shown on the same gel. This is an important piece of data to show that the interaction is not regulated by CDK9.

The result have been replaced by another experiment with the same outcome. All samples have been loaded on the same gel side-by-side (Figure 2b; see below)

5. Additionally, authors also need to address the effects of CSB on PAF1 localization relative to Rpb1 in the genome and to correlate such observation to the established PAF1-mediated H2B ubiquitylation in promoting RNAPII processivity (Wu et al, 2014). Do the loss of CSB impair PAF1 recruitment to the genome and hence, PAF1-mediated H2B ubiquitylation?

We have added new CHIP-seq data showing that PAF1 fails to show UV-induced repositioning in CSB-KO cells in the same set of genes that shows robust repositioning in WT cells (Figure 7a-e; see below). These findings show that CSB stimulates the UV-induced repositioning of PAF1 after UV.

We also included new CHIP-seq data showing strongly reduced binding of RNAPII at 8 hrs after UV irradiation in CSB-KO cells, which was not observed under similar conditions in WT cells. These findings suggest that CSB-KO cells through a combination of defective transcription-coupled repair, and their inability to both activate PAF1 and remove repressor ATF3 from TSS sites, show a very strong transcriptional repression at 8 hrs after UV irradiation (Figure 7f-h; see below).

Finally, we included new CHIP-seq data on ubiquitylated H2B (Ub-H2B) in WT and CSB-KO cells. We show that (page 11, line 15-27):

We detected clear Ub-H2B levels throughout genes in both unirradiated wild-type and CSB-KO cells (see Figure 7i for the KANSL1 gene). At 8 hrs after UV in wild-type cells, Ub-H2B levels were similar throughout early gene bodies (< 20 kb), with decreasing levels toward the end of longer genes (>100 kb). This is consistent with ongoing, but incomplete recovery of RNA synthesis particularly in long genes in wild-type cells at this time-point. Strikingly, in UV-irradiated CSB-KO cells, Ub-H2B levels progressively decreased much more rapidly within the first 20 kb of genes (see Figure 7i for the KANSL1 gene). We confirmed by immunofluorescence that absolute Ub-H2B levels were indeed more strongly decreased in CSB-KO cells (38%) compared to wild-type (63%) cells at 8 hrs after UV irradiation (Figure 7j). Metaplots of Ub-H2B averaged over 820 genes indeed showed a faster and more progressive loss of Ub-H2B deposition in CSB-KO cells in genes bodies after UV irradiation compared to wild-type cells (Figure 7k).

Figure 7

6. To authors should strengthen the assessment for the impact of PAF1 complex on the repair of CPD and 6,4-PP, as well as general cellular viability in addition to the TCR-UDS by using the AID system.

The repair of 6-4 PPs and CPD throughout the genome is dependent on global genome repair (GGR), with little or no contribution from transcription-coupled repair (TCR). The repair of these UV-induced photoproducts can therefore not serve as a measure of TCR.

To address if PAF1 is involved in GGR, we performed unscheduled DNA synthesis (UDS) experiment in PAF1-depleted cells using the AID system. In contrast to depletion of XPA, we find that loss of PAF1 is dispensable for GGR (Supplementary Figure S3c; below).

Measuring long-term viability in PAF1-depleted cells is not possible because PAF1 is an essential gene. We attempted to generate PAF1-KO cells, but were unsuccessful, which is why we generated PAF1-AID knockins instead. Short-term (48 hr) experiments after PAF1-depletion are perfectly possible in these cells (as also noted in *Hou et al. PNAS. 2019* using a similar approach), but clonogenic survival assays and long-term viability assays are, unfortunately, not possible in PAF1-depleted cells.

7. To authors should examine the impact of PAF1 on the genome localization and repositioning of RNAPII pSer2 and, in particular, pSer5 post UV. Metaplot data in Fig S4.D showed reduction RNAPII upstream of TSS in PAF-AID14 in addition to the loss of RNAPII in the proximal-promoter region. Recent study however reported that reduction of RNAPII elongation speed due to PAF1 depletion caused accumulation of pSer5 in the first 20 to 30 kb in the gene body (Hou et al, 2019).

Our ChIP-seq experiments using a pan-RPB1 antibody (recognizing both ser2-RPB1 and ser5-RPB1) largely confirm the results from the Dynlacht lab (Hou et al. PNAS. 2019), which were obtained in mouse cells. There is an increase in RNAPII reads in the first 2 – 3 kb compared to controls, which is likely due to slower RNAPII elongation. We have re-analysed the data from the Dynlacht paper, and the results are in line with our findings using a pan-RNAPII antibody (Supplementary Figure 7a, b; below).

Please note that we have recently shown that the pan-RPB1 antibody we have used in this study gives a signal close to the promoter, which overlaps closely with the Ser5-RPB1 antibody. However, a specific Ser2-RPB1 antibody gives no signal at the promoter, because Ser2 phosphorylation occurs after release from the promoter-proximal pause site (Nakazawa et al., 2020, Cell). The strong promoter peak seen in the ser2-RPB1 dataset from Hou et al. PNAS. 2019 demonstrates that the antibody used for these ChIP experiments was not specific for Ser2-RPB1, but likely recognizes Ser5-RPB1 as well.

In conclusion, we believe there is no discrepancy between the Dynlacht and our current study.

REVIEWERS' COMMENTS

Reviewer #2 (Remarks to the Author):

Authors have done a lot of work to address my comments. I found the manuscript now significantly improved and providing important insights into the role of PAF1C in the regulation of transcription in response to UV stress. I recommend the publication of the manuscript.